# Oligonuclear Actinoid Complexes with Schiff Bases as Ligands—Older Achievements and Recent Progress

**DOI:** 10.3390/ijms21020555

**Published:** 2020-01-15

**Authors:** Sokratis T. Tsantis, Demetrios I. Tzimopoulos, Malgorzata Holynska, Spyros P. Perlepes

**Affiliations:** 1Department of Chemistry, University of Patras, 265 04 Patras, Greece; sokratis.t.tsantis@gmail.com; 2Department of Chemistry, Aristotle University of Thessaloniki, 541 24 Thessaloniki, Greece; 3Department of Chemistry, Philipps University Marburg, Hans-Meerwein Strasse, 350 43 Marburg, Germany; 4Institute of Chemical Engineering Sciences, Foundation for Research and Technology-Hellas (FORTH/ICE-HT), Platani, P.O. Box 1414, 265 04 Patras, Greece

**Keywords:** actinoids, coordination chemistry, reactivity, Schiff-base complexes, structural studies, synthetic inorganic chemistry

## Abstract

Even 155 years after their first synthesis, Schiff bases continue to surprise inorganic chemists. Schiff-base ligands have played a major role in the development of modern coordination chemistry because of their relevance to a number of interdisciplinary research fields. The chemistry, properties and applications of transition metal and lanthanoid complexes with Schiff-base ligands are now quite mature. On the contrary, the coordination chemistry of Schiff bases with actinoid (5f-metal) ions is an emerging area, and impressive research discoveries have appeared in the last 10 years or so. The chemistry of actinoid ions continues to attract the intense interest of many inorganic groups around the world. Important scientific challenges are the understanding the basic chemistry associated with handling and recycling of nuclear materials; investigating the redox properties of these elements and the formation of complexes with unusual metal oxidation states; discovering materials for the recovery of *trans*-{U^VI^O_2_}^2+^ from the oceans; elucidating and manipulating actinoid-element multiple bonds; discovering methods to carry out multi-electron reactions; and improving the 5f-metal ions’ potential for activation of small molecules. The study of 5f-metal complexes with Schiff-base ligands is a currently “hot” topic for a variety of reasons, including issues of synthetic inorganic chemistry, metalosupramolecular chemistry, homogeneous catalysis, separation strategies for nuclear fuel processing and nuclear waste management, bioinorganic and environmental chemistry, materials chemistry and theoretical chemistry. This almost-comprehensive review, covers aspects of synthetic chemistry, reactivity and the properties of dinuclear and oligonuclear actinoid complexes based on Schiff-base ligands. Our work focuses on the significant advances that have occurred since 2000, with special attention on recent developments. The review is divided into eight sections (chapters). After an introductory section describing the organization of the scientific information, Sections 2 and 3 deal with general information about Schiff bases and their coordination chemistry, and the chemistry of actinoids, respectively. Section 4 highlights the relevance of Schiff bases to actinoid chemistry. Sections 5–7 are the “main menu” of the scientific meal of this review. The discussion is arranged according the actinoid (only for Np, Th and U are Schiff-base complexes known). Sections 5 and 7 are further arranged into parts according to the oxidation states of Np and U, respectively, because the coordination chemistry of these metals is very much dependent on their oxidation state. In Section 8, some concluding comments are presented and a brief prognosis for the future is attempted.

## 1. Scope and Organization of this Review

In the last 10 years or so, there has been a renaissance in the area of the chemistry and properties of metal complexes with Schiff bases as ligands [1]. Actinoid (An) ions seem to play an important role in this research area. New reaction schemes, novel structural types and exciting properties have been discovered. This review covers the most important (a subjective opinion!) synthetic routes or strategies that lead to dinuclear and oligonuclear An complexes with Schiff-base ligands, together with a brief description of their structures and properties. Due to lack of space, mononuclear and mixed metal-An complexes (with the exception of few alkali metal-An complexes) will not be covered; the literature on such complexes is very extensive, such that another review would be needed! Information about An complexes with redox-active, i.e., non-innocent, Schiff-base ligands is limited, because the relevant literature is also vast. Our work here focuses on the significant advances that have occurred since 2000. However, important contributions before 2000 are also mentioned whenever we believe that this helps the flow of the text and it is helpful for the reader. It should be emphasized at the outset that this report is not a fully comprehensive review. It aims to provide a taste for the subject as well as a critical examination of the current state of the topic, with an eye toward the future developments. Thus, apologies are due to the colleagues and researchers whose excellent work will not be cited here.

As far as we are aware, this is a first attempt to summarize and discuss the synthetic and structural chemistry, the properties and the potential applications of dinuclear and oligonuclear An complexes based on Schiff-based ligands. The topic has been partly covered in some older, excellent reviews or book chapters dealing with the chemistry of actinoids [2,3,4,5,6,7,8,9,10] and the coordination chemistry of Schiff bases [11].

The content of the review is purely chemical and it is assumed that the reader has a basic knowledge of structural and physical inorganic chemistry, including magnetic and optical properties. To avoid long synthetic descriptions, balanced chemical equations (written using molecular and not ionic-formulae) are used. In the text, we try to explain the synthetic rationale and philosophy behind the reactions with emphasis on the choice of reactants (ligands and metal-containing starting materials). Structural figures and description of physical properties are confined to the minimum. The method that is used to describe the binding of ligands to An ions herein is the “Harris Notation” [12]. This already widely-accepted method, describes the coordination mode as X.Y_1_Y_2_Y_3_…Y_n_, where X is the total number of metal ions bound by the whole ligand, and each Y value refers to the number of metal sites attached to the different donor atoms. The order of Y groups follows the Cahn–Ingold–Prelog priority rules; therefore, for the great majority of ligands reported in this work, O comes before N; from time to time the traditional η/μ notation will be also used. For clarity purposes, the coordination modes of many ligands reported in this review are presented schematically.

Section 2 and Section 3, which deal with general information about Schiff bases and their coordination chemistry, and the chemistry of actinoids, respectively, provide a “hors d’oeuvre” of the review. Section 4 aims to highlight the interest of the scientific community in the chemistry and properties of An complexes with Schiff bases as ligands; this section necessarily includes topics related to mononuclear, oligonuclear and polymeric complexes. Section 2, Section 3 and Section 4 are long (this is unusual for introductory parts) because we wish to attract the interest of scientists who are not familiarized with the An chemistry and the coordination chemistry of Schiff bases. The next sections (Section 5, Section 6 and Section 7) are the “main menu” of our scientific meal. We have chosen to arrange our discussion of dinuclear and oligonuclear An complexes according to the metal. The sections for Np and U are further arranged into parts according to the metal oxidation state; this approach is warranted because the coordination chemistry of actinoids is strongly oxidation-state dependent.

This review can be considered as a continuation of the interest of our group in some aspects of An chemistry [13,14,15,16] and coordination chemistry of Schiff bases with 3d [17,18], 4f [19,20,21] and 5f-metal [15,16] ions, with emphasis on dinuclear and polynuclear complexes and their magnetic, optical and catalytic properties.

## 2. Schiff Bases and Their Coordination Chemistry

Schiff bases—two emotive words! When the Italian–German chemist Hugo Schiff (1834–1915) synthesized the first members of this class of compounds that contained an azomethine (-HC=N-) or imine (>C=N-) group in 1864, he could not predict the impact of such molecules in modern chemistry.

The typical synthesis is rather straightforward and involves the condensation of a carbonyl compound (aldehyde or ketone) with a primary amine, often in refluxing conditions and under azeotropic distillation [11]. The formation of Schiff bases, which is sometimes acid-catalyzed, is a reversible reaction that proceeds via a carbinolamine intermediate (Figure 1); thus, removal of water is required to move the reaction to the right and to achieve high yields. Spectroscopically, Schiff bases are characterized by the appearance of an IR band in the 1680–1600 cm^−1^ region, depending on the different substituents on the C and N atoms, which is due to the stretching vibration of the carbon–nitrogen double bond, *v*(C=N). Most Schiff bases contain other functional groups; e.g., -OH, -COOH, -SH and pyridyl groups. Those containing a phenol group may exhibit tautomerism between the phenol-imine and the keto-amine forms in solution, which is governed by an intramolecular hydrogen bond. The dominant tautomer depends on the nature of the carbonyl precursor and not on the stereochemistry of the molecule or the substituent on the N atom [22]. The position of the equilibrium is strongly affected by the solvent and can be followed by ^1^H NMR and UV spectroscopies, the former being a particularly powerful tool for the study of this phenomenon. Phenol-imine and keto-amine forms are also present in the solid state. The most useful technique for the identification of the existing form is single-crystal X-ray crystallography [22]. For example, a shortening in the carbon–oxygen bond length (from 1.28 to 1.26 Å) and the lengthening of the imine carbon-nitrogen bond distance (from ca. 1.31 to 1.33 Å) indicate the predominance of the quinoidal structure (keto-amine form). The C atom of the imine bond is partially positively charged and it can undergo nucleophilic attack. Thus, for Schiff bases derived from amines containing another nucleophilic group, e.g., -SH or -NH_2_, an intramolecular nucleophilic attack is possible, thereby leading to products containing 5 or/and 6-membered heterocyclic rings.

Schiff bases are important compounds in organic chemistry exhibiting a great variety of applications [11,23], such as their use as chemosensors, polymer stabilizers and intermediates in organic synthesis; as dyes and pigments in the food industry; as catalysis in the synthesis of covalent organic frameworks (COFs); and in photochromism and thermochromism; the latter includes a broad range of biological activities, for which the azomethine or imine group plays a critical, albeit not fully understood, role.

Due to their highly modular synthesis that allows the control over the nature of donor sites; denticity; chelating and/or bridging abilities; and their electronic properties and steric characteristics, Schiff bases have played a prominent role in the development of modern coordination chemistry [11,22,24,25,26,27,28,29,30,31,32,33,34]. This is the reason they are often referred to as “privileged ligands” [27]. Schiff bases have been shown to coordinate to most main group-, transition- and non-radioactive f-metal ions. In the first years after the second world war, research efforts were directed towards synthesis and fundamental characterization of metal complexes that seem rather simple today. To mention an example, it was regarded as very significant to synthesize complexes of the general formula [Co^III^(X-sal-NR)_3_], with X = 5-Br, 5-NO_2_ and R = *^i^*Pr, where X-Hsal-NR is the general family of Schiff bases shown in Figure 2a. It had been believed that the isolation of such compounds would be very difficult if not impossible, because of the steric constraints imposed by the bulky *^i^*Pr substituent. The late R.S. Nyholm, one of the pioneers of modern coordination chemistry, used to say to his students [24]: “One type of excellent synthetic accomplishment would be to prepare those compounds that have so far been believed, but not proven, to be non-existent!” One of the best known and studied families of Schiff-base ligands, which contributed in the renaissance of coordination chemistry, is the so-called “salen-type” [33] (Figure 2b), in which the “mother” molecule is bis(salicylaldehyde)ethylenediamine (X = H, R′ = H, Y = CH_2_CH_2_). This important family consists of acidic (two -OH groups), tetradentate (2N, 2O) ligands.

Metal complexes with Schiff bases as ligands continue to attract the intense interest of inorganic chemists due to their relevance to a number of interdisciplinary research fields, including bioinorganic chemistry, molecular magnetism, multifunctional molecular materials, photo- and electroluminescence, energetic materials, sensors, materials with non-linear optical properties, medicinal chemistry, homogeneous and heterogeneous catalysis and multielectron redox chemistry [1,11,18,19,20,21,22,24,25,26,27,28,29,30], among others. For example, in the field of catalysis, Schiff-base metal complexes can catalyze polymerization, ring-opening polymerization, oxidation, epoxidation, ring opening of epoxides, reduction of ketones, allylic alkylation, hydrosilation of acetophenones, hydrogen peroxide decomposition, Michael addition, annulation, carbonylation, Heck reactions, alanine benzylation, hydrocarbon amidation and aziridination, cyclopropanation, aldehyde silylcyanation, Diels-Alder and aldol condensation reactions, isomerization of norbornadiene to quadricyclone, addition reactions of cyanides to imines and desymmetrization reactions of *meso* compounds [26,27]. The rationale behind the great catalytic activity of Schiff-base metal complexes is that Schiff bases are able to stabilize many different metals at various oxidation states, thereby controlling the performances of metal ions in an enormous variety of useful catalytic transformations [27]. We finalize this section by providing an impressive example of the redox activity of tetradentate N,N,O,O Schiff bases with the objective of identifying new pathways to lanthanoid (Ln) multielectron redox transfer [29]. The chemical reduction with alkali metals of heteroleptic [Nd(salophen)X] (X = I, CF_3_SO_3_) and of a variety of homoleptic K[Ln(^R^salophen)_2_] (Ln = trivalent lanthanoid) complexes has resulted, respectively, in the dinuclear Nd(III) complex K_2_[Nd_2_(*cyclo*-salophen)(THF)_2_] and in a series of mononuclear Ln(III) complexes of general formula K_3_[Ln(bis-^R^salophen)] (R = H, Me, *^t^*Bu); the organic ligands involved are illustrated in Figure 3. Ligand reduction and C–C bond formation have been supported by single-crystal X-ray crystallography. NMR studies have demonstrated that the dinuclear Nd(III) complex can transfer four electron in the reaction with oxidizing agents, such as Ag(CF_3_SO_3_), through the breaking of the two C–C bonds. Electrochemical and reactivity studies of the mononuclear complexes K_3_[Ln(bis-^R^salophen)] have shown that they can behave as formal two-electron reductants, and that their oxidation potential can be tuned by changing the substituent on the ligand. These systems are the first examples of mononuclear Ln(III) complexes capable of two-electron transfer and the first dinuclear Ln(III) complex that can transfer four electrons from a single molecule/ion.

## 3. A General Overview of the Chemistry of Actinoids

The element actinium (Ac) is strictly a group 3 metal. However, its chemical characteristics being identical to Th-Lr has as a consequence, the classification of Ac with the actinoids. The IUPAC recommends the name actinoid and not actinide; this is because the ending “ide” is usually used for anions. Some information for the actinoids is provided in Table 1. Th and U are the only An metals which have naturally occurring isotopes. Studies of the inorganic and coordination chemistry of the elements with Z > 92 (the *transuranium* elements) are limited because specialized experimental techniques are required for handling the elements and their compounds. All the actinoids are unstable undergoing radioactive decay. The half-lives of the most abundant isotopes of thorium (^232^Th) and uranium (^238^U) are very long (*t*_1/2_ = 1.4 × 10^10^ and 4.6 × 10^10^ years, respectively) and their radioactivity can be neglected.

The systematic study of the chemistry of actinoids [2,8,33,34,35,36,37,38,39,40,41,42,43,44,45,46,47] started in 1895 when Becquerel discovered that U undergoes radioactive decay. The discovery of artificial radioactivity in 1934 set more experimental work alight because scientists wanted to prepare new elements that did not exist in nature, and the first man-made element, Np, was produced in 1940. Descriptive coordination chemistry played a significant role in this discovery period, because the chemical properties of the actinoids (stoichiometry of binary compounds, oxidation states, reactivity) were used to argue for their position in the Periodic Table of the Elements. A milestone in the chemistry of actinoids was the discovery of nuclear fission in 1938. ^235^U and ^239^Pu both undergo fission with slow neutrons, creating neutron chain reactions and making them suitable for the production of weapons in the context of the Manhattan Project. This stimulated large-scale separation chemistry, precipitation recovery processes and methods for the extraction of An ions from aqueous into nonaqueous solutions by the use of various extractants.

The existence of at least two oxidation states for nearly all the actinoids implies that the successive ionization energies differ little. For the higher oxidation states, there is significant covalent character in their chemical bonding. The great radial extent and energetic availability of the 5f and 6d orbitals result in increased interactions with ligand-based orbitals; another reason is that the energy separations between 5f, 6d, 7s and 7p atomic orbitals are generally small so that appropriate valence states for covalent bonding can be attained. There are some distinct differences between the chemistry of the early actinoids (Th-Pu) and that of the later ones (Am-Lr). The early metals are much more readily available (either as products of nuclear materials and fuel production or from natural ores), whereas the later elements are rarer and only available in very limited quantities. The early actinoids exhibit the greatest range of accessible oxidation states; this follows from the lowering in energy of the 5f orbitals on crossing the period. Isotopes of the early actinoids have longer half-lives, reducing the possibility of self-radiolysis and yielding more stable products. Last, but not least, broader interest exists in the technology of the early actinoids because these are more useful for energy production, and, unfortunately, for the production of nuclear weapons. This implies that there is currently intense interest in the environmental chemistry of early actinoids. For all these reasons, the coordination and organometallic chemistry of these elements is much more developed than that of the later ones.

The ions of the actinoids have rather large ionic radii (Table 1) and high coordination numbers (normally 8–10), and are thus common; even higher coordination numbers have been found (Figure 4). Preferences between different coordination numbers and geometries are usually controlled by steric effects. Ligands are rather labile, and kinetic barriers for reactivity are not large. Ionic radii across the series decrease, and therefore, accessible coordination numbers may decrease from left to the right for a given oxidation state with the same ligand.

Another aspect of the actinoid chemistry is that metals in the IV, V and VI oxidation states usually form bonds to two “yl” O atoms, forming linear dioxo actinyl species {OAnO}^n+^ (*n* = 0–2). The presence of the two yl O atoms in high-valence An coordination polyhedra ensures that their oxido clusters present interesting structural types and topologies. Linear *trans*-dioxo (or dioxido) actinyl cations arise form the interaction of An and O orbitals. For example, in the *trans*-{U^VI^O_2_}^2+^ [uranyl(VI)] ion, which is the dominant form of uranium in the environment, these U-O_yl_ bonds are formally triple bonds consisting of one σ and two π bonds between 2p orbitals on the oxo groups and hybrid orbitals (5f and 6d) of uranium.

The solution and solid-state structural chemistry of hydrated actinoid “salts” and their hydrolysis and condensation products are deeply studied and well understood topics. For several decades, most studies with early An complexes have involved aqueous systems of relevance to nuclear fuel processes. Under such conditions, one may operate at, or close to, thermodynamic sinks; or the processes can be supported by organic ligands that are compatible with water. Under strictly anhydrous conditions, such restrictions are relaxed and give the chance to prepare and study complexes that would not normally exist in a normal laboratory atmosphere. The non-aqueous early An chemistry is currently a “hot” research topic providing access to “exotic” compounds and exciting properties.

The chemistry of actinoids is both shared with and different to that of lanthanoids (Ln). Their ions are considered as “hard” acids (according to the HSAB model) and form stable complexes with ligands that possess donor atoms that are “hard” bases; i.e., O; the Ln and An ions are also susceptible to hydrolysis. Both ions show a contraction of ionic radius as the atomic number increases and an increasing reluctance to exhibit higher oxidation states later in the series. Most compounds are paramagnetic, but the electron spin-nuclear spin relaxation times often give rise to well-resolved NMR spectra and do not permit the observation of EPR spectra, except at low temperatures. The metals display more than one accessible oxidation state, and one-electron redox chemistry is usual. Due to their similar and chemical properties, the partitioning of An and Ln ions (which is very important in nuclear industry) is the most challenging hydrometallurgical separation known. Concerning the differences, the chemistry of An elements is much more diverse than that of their Ln counterparts. The covalent character of the chemical bonding in An compounds (vide supra) is more pronounced than in Ln compounds. While the electronic structure of Ln complexes is dominated by spin-orbit coupling and electron-electron repulsions, that of An complexes is often influenced by ligand-field effects, giving absorption spectra; absorptions due to 5f–5f transition are weak, but they are more intense, broader and considerably more dependent on the ligands present than those attributed to 4f–4f transitions. The interpretation of the electronic spectra in An complexes is becoming complicated by the large spin-orbital coupling constants which are almost double those of lanthanoids. The magnetic properties of An compounds show a general similarity to those of Ln compounds in the variation of the effective magnetic moment with the number of unpaired electrons. However, the values for Ln and An species containing the same number of f electrons—Np(VI) and Ce(III), Np(V) and Pr(III), Np(IV) and Nd(III), etc.—are lower for the An complexes, suggesting partial quenching of the orbital contribution by effects of the crystal field.

The chemistry of actinoids is currently attracting the intense interest of many inorganic chemistry groups around the world. Important scientific challenges for research are handling and recycling nuclear materials, discovering efficient materials for the recovery of {U^VI^O_2_}^2+^ from seawater, investigating the redox properties of these elements, elucidating and manipulating An-elements’ multiple-bond interactions, finding methods to accomplish multi-electron reactions related to organometallic transformations and improving the 5f-elements’ potential in small-molecule activation.

## 4. Scientific Interest in the Chemistry of Actinoid-Schiff Base Complexes

The study of An complexes with Schiff bases as ligands is an important research theme for a variety of reasons.

From the synthetic inorganic chemistry viewpoint, polydentate Schiff-base ligands offer an ideal platform for stabilizing and promoting the reactivity of the An ions. For example, planar pentadentate Schiff bases offer a well-suited binding pocket to accommodate the size and the preference for a pentagonal bipyramidal coordination geometry of the U^VI^ center in the *trans*-{U^VI^O_2_}^2+^ ion; this chemistry remains poorly investigated [11]. This cation is also an efficient template for condensation of diketones/dialdehydes and diamines, resulting in uranyl complexes containing macrocyclic Schiff-base ligands [5,10]. The main problem for the synthesis of macrocyclic ligands is associated with the appropriate orientation of reactive sites, which should give intramolecular (cyclic) rather than intermolecular (acyclic) products. Judicious location of donor atoms and rational choice of the metal ion can provide synthetic control over such a cyclization process [30]. Template reactions are not limited to those involving a single organic component. They may occur with two open chain precursors that contain mutually reactive groups. In the uncontrolled reaction, the normal product is a polymeric compound. Coordination of one reactant to a metal ion holds the reactive sites in the correct conformation for reaction with the second organic component. Thus, once the first reaction has occurred, the intermediate complex is correctly oriented for intramolecular reaction. The macrocyclic complexes can have [1 + 1], [2 + 2] or other stoichiometries [30]. A typical example [5,10] of the use of the uranyl cation as a template for [2 + 2] Schiff-base condensation is illustrated in Figure 5. The *trans*-{U^VI^O_2_}^2+^ species stabilizes the product, because-in addition to the preference for a pentagonal bipyramidal geometry mentioned above—it forms complexes with a hexagonal bipyramidal coordination.

Another interesting area in the chemistry of actinoid-Schiff base complexes is the ability of neutral uranyl complexes with tetradentate chelating “salen-type” ligands (Figure 2b) to behave as receptors for anions. The design and synthesis of selective anion receptors is an important topic in supramolecular chemistry. In nature, the selective complexation of anions takes place by hydrogen bonds; for example, the selective recognition of phosphate and sulfate ions in biological systems by transport receptor proteins is well documented. An early example in this areas was the reaction of the “naked” salophen [UO_2_(^MeO^salophen)], which contains one vacant position for guest coordination, with one equiv. of (*^n^*Bu_4_N)(H_2_PO_4_) in MeCN that gives the orange complex (*^n^*Bu_4_N)[UO_2_(^MeO^salophen)(H_2_PO_4_)] [48]; the structural formula of ^MeO^salphen^2−^ is analogous with that of ^Me^salophen^2−^ (Figure 3), the only difference being the identity of R which is Me in ^Me^salophen^2−^ and OMe in ^MeO^salophen^2−^. In the complex anion (Figure 6), the U^VI^ atom has an approximate pentagonal bipyramidal coordination, with the two uranyl oxygen atoms in axial positions. In the equatorial plane, in addition to coordination by the two O and two N atoms of the tetradentate chelating ^MeO^salophen^2−^ ligand, an O atom from H_2_PO_4_^−^ anion occupies the fifth equatorial site. In the solid state, the complex anions are arranged in centrosymmetric pairs by strong H bonds. This early work inspired studies on anion-facilitated transport through supported liquid membranes and on H_2_PO_4_^−^ sensors [6].

Some actinoid-Schiff base complexes are also useful homogeneous catalysts for important reactions. The Lewis acid character of uranyl complexes has been particularly exploited [49,50,51]. Following earlier studies on the use of the uranyl(VI)-tetradentate Schiff base unit as an electrophilic catalyst of acyl transfer reactions [49], the groups of Reinhoudt and Mandolini [50,51] explored the potential of complexes [UO_2_(salophen)] and [UO_2_(^Ar^salophen)] to act as efficient catalysts for 1,4-thiol addition reactions, where ^Ar^salophen^2−^ is the dianionic ligand with R = phenyl ring and R′ = H (Figure 3). The catalyst design was based on the well-known property of salophen-uranyl complexes to bind another anionic or polar neutral ligand in an equatorial coordination site to give pentagonal bipyramidal complexes. They selected the model reaction of thiophenol with 2-cyclopenten-1-one in the presence of Et_3_N in CHCl_3_ at 25 °C (Figure 7). The uncatalyzed reaction (to be strictly correct, the reaction is Et_3_N-catalyzed) proceeds with *t*_1/2_ of 162 min. Upon the addition of catalytic amounts of [UO_2_(salophen)], the *t*_1/2_ is lowered to ≈13 min, while [UO_2_(^Ar^salophen)] demonstrated a *t*_1/2_ of 3.8 min; the TOF values at *t*_1/2_ for the catalysts are 1.43 and 0.63 s^−1^, respectively. The better performance of [UO_2_(^Ar^salophen)] relative to [UO_2_(salophen)] is probably due to attractive van der Waal’s forces between the 2-cyclopenten-1-one and the phenyl groups of the catalyst [6]. Concerning the mechanism of the catalysis, it has been proposed that an enone-uranyl(VI)-salophen complex forms via coordination of the enone carbonyl group to the uranyl(VI) center. The “activated” enone then reacts with a base-thiol adduct [6]. 

The development of new An separation strategies is of major importance if many of the challenges related to used nuclear fuel processing and nuclear waste management are to be addressed [7]. In the first step of a typical fuel separation, the conventional PUREX (plutonium and uranium reduction extraction)-based solvent extraction process is used; this relies on the extraction of uranyl nitrate by tri-*n*-butyl phosphate into an organic phase [52]. However, keeping the bulk of the uranium out of the solvent phase could reduce the volume of solvent required for the nuclear fuel processing operations. Thus, extraction of the minor components (e.g., Ln(III) ions and transuranium elements) from the uranium-containing phase would be highly preferable. The proposed separation methods of minor An and Ln ions often employ organophosphorus reagents. These extractants do not display the desired large selectivity between An(III) and Ln(III) species, because of the similarities in the chemical properties of these metal ions. Therefore, these extractants are frequently applied in combination or in sequence with soft (HSAB) donors to achieve Ln(III)/An(III) separations. The idea of selectively retaining metal ions in an aqueous phase during solvent process appears very promising for Ln(III)/An(III) separation techniques such as TALSPEAK (trivalent actinoid-lanthanoid separation by phosphorous reagent extraction from aqueous “komplexes”) and Reverse TALSPEAK, and the most recent, innovative SANEX (selective actinoid extraction) process concepts [53,54]. On the other hand, the early An elements possess higher oxidation states available (Table 1) and are frequently exist as *trans* dioxo cations. Having this structural configuration, the An elements behave very differently from trivalent lanthanoids. The stabilities of these oxidation states are a challenge for some transuranium actinoids under acidic conditions. However, studies have suggested that such oxidation states may be stabilized by coordination of polydentate chelating ligands in the equatorial plane of the actinyl cation. For example, recent results have indicated that Schiff bases may stabilize U(V) [55]. The ability of tetra- and pentadentate Schiff-base ligands to accommodate the steric demands of *trans*-{U^VI^O_2_}^2+^ has led to intense interest in their application as extractants for this cation. In an excellent, relatively recent study, Hawkins’ group evaluated the water-soluble tetradentate *N*,*N*′-bis-(5-sulfonatosalicylidene)- ethylenediamine (H_2_salen-SO_3_^2−^, Figure 8) and pentadentate *N*,*N*′-bis-(5-sulfonatosalicylidene)- diethylenetriamine (H_2_saldiene-SO_3_^2−^, Figure 8) Schiff bases for their ability to form complexes with UO_2_^2+^ and Ln(III) ions in aqueous solution [7,56]. U(VI) could be selectively complexed and retained in the aqueous phase, whereas the representative trivalent lanthanide Eu(III) was extracted by bis(2-ethylhexyl)phosphoric acid in toluene. Such results hold promise for the application of water-soluble polydentate Schiff-base ligands in the development of novel Ln/An separation schemes.

In bioinorganic chemistry, thermodynamically stable complexes of An ions with chelating Schiff bases derived from biologically relevant molecules can be considered models for the therapeutic development of clinical chelators that can be used for the detoxification of An ions. For example, the Schiff base *N*,*N*′-bis(pyridoxylideneimine)ethylene, H_2_(pyr)_2_en (Figure 9), derived from the condensation of pyridoxal with ethylenediamine, forms the stable complex [Th^IV^{(pyr)_2_en}_2_(H_2_O)] [57]. This pyridoxal-containing ligand can combine the metabolic potential of the vitamin B family with the chemical ability to form stable chelates.

Some An ion-selective chromogenic sensors are based on the complexation ability of polydentate Schiff bases. For example, the intake of thorium in humans has acute toxicological effects and can cause lung, pancreatic and bone cancers. Because of the widespread usage of this metal and its compounds, current attention is being paid to developing highly selective chemosensors for the detection/determination of thorium with low detection limits, wide pH ranges and minimal interference from potential competing ions. A new Th^IV^-selective chromogenic chemosensor was based on the Schiff-base ligands derived from the condensation of 1,10-phenanthroline-2,9-dicarbohydrazide with 2-hydroxynaphthaldehyde. The great sensing performance of the polydentate ligand towards Th^IV^ was investigated in solution and paper strips loaded with Schiff base using spectrophotometric and colorimetric techniques [58].

Actinoid-Schiff base complexes are useful in the knowledge and grasp of fundamental An bonding trends and coordination preferences [59,60], Such complexes have thus attracted the intense interest of theoretical chemists, who have been trying to understand the bonding and the electronic structure trends across the 5f-metal series and how the changes in ligand binding may impact speciation and separation selectivity to different metal ions. For example, there are only few comparisons of isostructural molecular complexes with the metals in the IV oxidation state. Albrecht–Schmitt and co-workers prepared and fully characterized a series of homoleptic transition-metal, lanthanoid and actinoid complexes of the general formula [ML_2_] (M = Zr, Hf, Ce, Th, U, Pu), where L^2−^ is the dianion of the salophen-type ligand *N*,*N*′-bis[(4,4′-diethylamino)- salicylidene]-1,2-phenylenediamine (Figure 10left). These complexes offer the opportunity for the comparison of isostructural species containing metals from different positions of the Periodic Table in the tetravalent oxidation state. The metal ions are 8-coordinate and feature a co-facial ligand geometry (Figure 10right). Time-dependent DFT and other quantum mechanical methods were used to investigate bonding differences between the complexes, and the calculated absorption spectra are in good agreement with the experimental ones. The computational study also revealed that the U(IV) and Pu(IV) complexes have more covalent character in their bonding than that found with the other metal ions; [CeL_2_] showed increased covalent behavior compared to [ThL_2_].

Schiff bases have also played an important role in the reduction and functionalization chemistry of {U^VI^O_2_}^2+^ [9]. As mentioned in Section 3, the uranyl(VI) dication is thermodynamically stable and rather inert to chemical functionalization in the normal laboratory atmosphere. It has been recently discovered that under anaerobic conditions, compounds containing the uranyl(V), {U^V^O_2_}^+^ and U(IV) dioxo {U^IV^O_2_} species can be synthesized through reductive functionalization processes. The {U^V^O_2_}^+^ cation disproportionates into {U^IV^O_2_}^2+^ and {U^IV^O_2_} under anaerobic conditions, with the latter being insoluble in H_2_O. Functionalization can also occur in uranyl(V) complexes without reduction. There is an increasing interest in the study of uranyl(V) complexes because the more Lewis-basic oxo groups can bridge other metal ions; these interactions are frequently referred to as cation–cation interactions (CCIs). This phenomenon is extremely rare in uranyl(VI) chemistry, but rather common in heavier actinyl, i.e., Np and Pu, chemistry. Through the formation of oxo-bridged complexes, U^V^ can disrupt nuclear waste separations; e.g., in the PUREX process; thus, scientists have been trying to understand such interactions in depth. For example, the reaction of the readily available uranyl(V) starting material {[U^V^O_2_(py)_5_KI_2_(py)_2_]}_n_ with the Schiff-base salt K_2_(*^t^*^Bu^salophen) (Figure 3) in pyridine (py), affords a mixture of disproportionation U^VI^ + U^IV^ products. However, when the bulkier Schiff-base salt K_2_(*^t^*^Bu^salophen) (Figure 3) is employed, the polymeric compound {[KU^V^O_2_(*^t^*^Bu^salophen)(py)]}_n_ is obtained, which is stable toward disproportionation for almost one month in py, DMSO and toluene (Figure 11). This highlights the important effect that increasing the steric protection of the salophen^2−^ ligand framework increases the stability of uranyl(V) species. In accordance with this, the product does not react with H_2_O to result in ligand protonation and uranyl(V) oxidation. The reaction illustrated in Figure 11 is a nice example of further functionalization of an uranyl(V) complex that retains the U(V) oxidation state [9].

## 5. Oligonuclear Schiff-Base Complexes with Actinoids Other Than Thorium and Uranium

As mentioned in Section 3, studies on the chemistry of most An elements are very limited due to their radioactivity. A mononuclear Pu(IV) complex with a N,O-bidentate chelating β-ketoiminate ligand has been structurally characterized [61]. Computational studies on this and analogous U(IV) complexes reveal significant metal–ligand interaction differences between U(IV) and Pu(IV) bonding. The mononuclear compound [Np^V^O_2_(salen)(MeOH)] is also known [62].

### 5.1. Tetranuclear Neptunyl(V) Clusters Supported by Salen^2−^ Ligands

Np is a reactive metal which quickly tarnishes in air. Its reaction with dilute acids produces H_2_, but the metal is not attacked by alkali. ^237^Np (*t*_1/2_ = 2.144 × 10^6^ years) is an active α, i.e., ^4^_2_He^2+^, emitting radionuclide which is generated in nuclear reactors. Its limited coordination chemistry is related to the development of effective minor An (as ^237^Np) separation processes; such processes are crucial to reducing the radiotoxicity of nuclear waste streams; otherwise, this isotope will contribute a lot to radiation from these wastes during their storage for many years. The neptunyl(V) cation, {Np^V^O_2_}^+^, is relevant in nuclear reprocessing technology and in the environmental issues arising from this technology. Because of the low positive charge of {NpO_2_}^+^, the design of ligands for the selective separation of this actinyl ion from spent nuclear fuel is not easy.


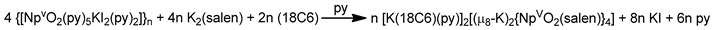
(1)

The reaction of {[Np^V^O_2_(py)_5_KI_2_(py)_2_]}_n_ and K_2_(salen) in py under inert and anhydrous conditions, in the presence of 18-crown-6(18C6) leads [63] to cluster as [K(18C6)(py)]_2_[(μ_8_-Κ)_2_{Np^V^O_2_(salen)}_4_] (**1**) at a 56% yield (Equation (1)); the ligand H_2_salen is shown in Figure 2b (X = H, R′ = H, Y = CH_2_CH_2_).

The structure of the cluster anion [(μ_8_-Κ)_2_{Np^V^O_2_(salen)}_4_]^2−^ (Figure 12a) consists of four neptunyl(V) cations, each coordinated in the equatorial plane by a tetradentate ligand through two Np-O, two Np–N bonds and one bridging O atom from an adjacent {NpO_2_(salen)}^−^ unit [63]. Thus, the salen^2−^ ion behaves as a 2.2111 ligand (Figure 12b). Two K^I^ centers reside above and below the mean plane defined by the four An ions; each alkali cation is connected to four salen^2−^ O atoms and four neptunyl(V) oxido groups, the mean K-O_yl_ distance being ≈3 Å. Each neptunyl(V) unit is almost linear with a mean O-Np-O angle of ≈178°. ^1^H NMR studies in pyridine-d_5_ suggest that the complex retains its tetranuclear nature in solution. The influence of the alkali cation on the chemical and structural identity of the product was investigated by the 1:1 reaction of {[Np^V^O_2_(py)_5_LiI_2_(py)_2_]}_n_ with K_2_(salen) in py, which led to the isolation of [Li(py)_2_]_2_[(μ_8_-Κ)_2_{Np^V^O_2_(salen)}_4_] (**2**). Although the crystal structure of the product was not of good quality, the data showed the presence of the same ion [(μ_8_-Κ)_2_{Np^V^O_2_(salen)}_4_]^2−^ found in **1** with two additional interactions between the Li^I^ ions and the neptunyl(V) oxido atoms [63]. The major chemical message of this work is that the use of the tetradentate Schiff base, which imposes the appropriate steric constraints around the {Np^V^O_2_}^+^ equatorial plane, permits the operation of CCIs, yielding a discrete tetranuclear cluster.

### 5.2. Unique Np(III)/U(VI) Complexes

In an attempt to study the reduction and functionalization chemistry of {U^VI^O_2_}^2+^, the groups of Arnold and Love performed [64] the reactions of [U^VI^O_2_(H_2_^R^L^1^)(THF)] and [Np^III^(Cp)_3_] in THF, where (H_2_^R^L^1^)^2−^ are the dianions of “Pacman” Schiff-base polypyrrolic macrocycles H_4_^Me^L^1^ (R = Me in Figure 13) and H_4_^Et^L^1^ (R = Et in Figure 13), and Cp is the cyclopentadienyl ligand (Equation (2)). A noticeable color change from greenish (the color of the {U^VI^O_2_}^2+^-containing starting material) to red-brown was observed, and dark red crystals of [Np^III^(Cp)_3_OU^VI^O(H_2_^R^L^1^)(THF)] (R = Me, **3a**; R = Et, **3b**) were isolated at yields of ≈30%. The complexes are highly air-sensitive, but in general, the octamethyl ligand derivative is more easily isolated. Compounds **3** are the only molecular heterometallic neptunium-uranium complexes reported to date. The oxidation states of Np and U are ambiguous. While certain X-ray, ^1^H NMR and IR data suggest that a U^VI^→U^V^ reduction occurs in the products, the Np-O_exo_ bond distances and SQUID magnetometry results suggest that the two complexes can be best described as donor–acceptor uranyl oxido-bridged Np^III^/U^VI^ complexes with only partial electron transfer from Np^III^ to U^VI^ occurring. The later evidence [9,64] is further supported by DFT calculations.


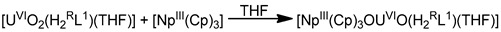
(2)

## 6. Dinuclear and Oligonuclear Thorium(IV)—Schiff Base Complexes

Thorium is named after Thor, the Scandinavian god of war and thunder [68]. It is rather stable in the atmosphere, but reacts slowly with H_2_O, and rapidly with steam and dilute hydrochloric acid. Upon heating, the metal reacts with H_2_ to give ThH_2_; halogens to yield ThX_4_; and with C and N to give carbides and nitrides, respectively. It forms alloys with several metals; e.g., ZnTh_2_ and CuTh_2_. The chemistry of thorium largely concerns Th(IV). In an aqueous solution, there is no evidence for other oxidation states, the E° value for the Th^4+^/Th couple being −1.19 V [34]. There are only a handful of crystallographically characterized examples of Th(III) in the literature [69], and most contain cyclopentadienyl ancillary ligands. The first Th^IV^/Th^III^ redox couple values have been recently determined experimentally employing CV measurements, which have been facilitated by the use of (BPh_4_)(*^n^*Bu_4_N) as a supporting electrolyte in THF. Th(IV) and Th(III) metallocene compounds were studied, and their redox couple values range from −2.96 V to −3.32 V versus [Fe(Cp)_2_]^+/0^. Because of its large size, coordination complexes of Th(IV) exhibit high coordination numbers (e.g., Figure 4) and hard donors are preferred. Lower coordination numbers can be stabilized by amido or aryloxido ligands, and the isolation of the first square planar Th(III) complexes (and as a matter of fact the first examples of this geometry for f element complexes) has been just described [70].

There has been a renaissance in the chemistry of Th(IV) complexes in the last few years. This is mainly due to the potential of Th for the next generation of nuclear fuel, as the development of liquid-fluoride thorium reactors is close to commercialization [40,71]. Since the main Th production mineral is monazite, which also contains Ln^III^ ions, there is an urgent need for the discovery of suitable organic ligands and efficient solid adsorbents for the selective extraction of Th^4+^ from 4f-metal ions [72,73]. Other research topics of interest in the Th(IV) complexes field are studying the solutions and solid products arising from Th(IV) hydrolysis [35]; the investigation of Th(IV)-peroxide chemistry [74]; the incorporation of this diamagnetic metal ion into isostructural heterometallic d/U^IV^ clusters with the goal of elucidating the d–d magnetic exchange interactions [75]; the structural characterization of complexes with very high coordination numbers and novel polyhedra [41]; the progress in complexes with Th(IV)-ligand multiple bonds [76,77]; the stabilization of novel secondary building units in Th(IV) metal-organic frameworks [78]; the study of the electronic and thermodynamic properties of Th(IV)/An(Z) (Z = various oxidation states) and An(Z)/Th(IV)/3d-metal ion MOFs (Metal-Organic Frameworks) with “structural memory” [79]; and the in-depth investigation of Th(IV) interactions with ionic liquids to model aspects of the An ions extraction from radioactive feeds [42].

Concerning the importance of Th(IV)-Schiff base coordination chemistry, we have already mentioned (Section 4) that it is associated with some bioinorganic chemistry aspects [57]; i.e., the development of colorimetric detection of this metal ion and applications in real-time samples [58] and the understanding of fundamental An bonding trends [59,60]. Additionally, Schiff-base complexes of Th(IV) are useful for the construction of on/off sensors for this metal ion to exploit the ligand-based fluorescence of its complexes [80].

The to-date structurally characterized dinuclear and oligonuclear Th(IV) complexes with Schiff-base ligands and the coordination modes of the latter are listed in Table 2. The structural formulae of the Schiff bases (presented in their neutral forms) and their abbreviations (used in this review) are illustrated in Figure 13. The coordination modes of the ligands in the complexes are shown in Figure 14.

### 6.1. Dinuclear Thorium(IV) Complexes

The 1:1 reaction between Th(NO_3_)_4_·5H_2_O and the planar ligand 2,9-diformyl-1,10- phenanthrolinedisemicarbazone (H_2_L^2^, Figure 13) in EtOH/H_2_O gives complex [Th_2_O(NO_3_)_2_(H_2_L^2^)_2_(H_2_O)_2_](NO_3_)_4_ (**4**) [65]. The H_2_L^2^ molecule can be also considered as a substituted hydrazone. The two Th^IV^ centers in the dinuclear cation are bridged by a μ-O^2−^ group. A hexadentate chelating 1.1111110000 neutral H_2_L^2^ ligand (Figure 14), a bidentate chelating nitrato group and a H_2_O molecule complete the 10 coordination sites of each Th^IV^. The semicarbazone side chains of each ligand are twisted slightly relative to the phenanthroline plane. The coordination polyhedra of the two metal ions are highly irregular and the authors describe them as 1-6-3 polyhedra.

The reaction of the potentially pentadentate compartmental Schiff base H_3_L^3^ (Figure 13), derived by condensation of 2,6-diformyl-4-chlorophenol and two equiv. of *o*-aminophenol, with Th(NO_3_)_4_·5H_2_O and Mg(O_2_CMe)_2_·4H_2_O in MeOH/DMF gave complex [Mg(H_2_O)_6_][Th_2_L^3^)_3_]_2_ (**5**); Equation (3) [66]. In each [Th_2_(L^3^)_3_]^−^ anion, the two Th^IV^ ions are bridged by three phenoxido atoms of the trianionic pentadentate 2.21111 ligands (Figure 14 and Figure 15). The Th^IV^ centers are 9-coordinate with a slightly distorted tricapped trigonal prismatic geometry. The pentadentate ligands are not planar, the two terminal phenyl rings being inclined with respect to the central ring of each ligand by 36°–54°. The [Mg(H_2_O)_6_]^2+^ cation is octahedral.


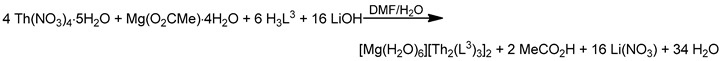
(3)

The 1:1 reaction between Th(NO_3_)_4_·5H_2_O and the polydentate ligand H_2_L^4^ (Figure 13) in EtOH/H_2_O yields the hydroxo-bridged dinuclear complex [Th_2_(OH)_2_(NO_3_)_2_(H_2_L^4^)_2_(H_2_O)_2_](NO_3_)_4_ (**6**) [67] (Equation (4)). The Th^IV^ centers in the centrosymmetric dinuclear cation (Figure 16) are bridged by the two hydroxido groups. A pentadentate 1.111110000 chelating H_2_L^4^ ligand (Figure 14), one bidentate chelating nitrato group (1.110) and one terminal H_2_O molecule complete 10-coordination at each metal ion. The best description of the coordination polyhedra is that of a distorted bicapped square antiprism. The dihedral angles between the aromatic ring and the side arms are ≈21°. Such distortions are possible only when the ligand is neutral; when deprotonation of the semicarbazone -NH- groups occurs, conjugation leads to ligand planarity.


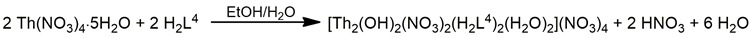
(4)

### 6.2. Two Tetranuclear Thorium(IV) Complexes

The highest known nuclearity in Th(IV)-Schiff base chemistry is four and the oligonuclear complexes come from our laboratories [16]. The reactions between Th(NO_3_)_4_·5H_2_O and *N*-salicylidene-*o*-aminophenol (H_2_L^5^ in Figure 13) and *N*-salicylidene-*o*-amino-4-methylphenol (H_2_L^6^ in Figure 13) in MeCN (without or with the addition of an external base) led to yellowish orange complexes [Th_4_O(NO_3_)_2_(HL^5^)_2_(L^5^)_5_] (**7**) and [Th_4_O(NO_3_)_2_(HL^6^)_2_(L^6^)_5_] (**8**), respectively, with ≈40% yields (Equation (5)). The molecular structures of the complexes are similar. The four Th^IV^ centers are arranged at the vertexes of a distorted tetrahedron with a μ_4_ (4.4)-oxido group bonded to each actinoid ion, thereby creating the extremely rare {Th_4_(μ_4_-O)} unit. The Th^IV^ ions are held together by one 3.221 doubly deprotonated ligand, four 2.211 doubly deprotonated ligands and two 3.21 singly deprotonated ligands (Figure 14). The H atoms of one of the acidic -OH groups of each singly deprotonated group were found on the imine nitrogen atom, and this blocks its coordination. The Th^IV^ centers adopt coordination numbers of 8, 9 and 10 with a total of four different coordination polyhedra (triangular dodecahedron, muffin, biaugmented trigonal prism and sphenocorona). The core of the complexes appears to be {Th^IV^_4_(μ_4_-O)(μ-OR″)_8_} (Figure 17). A plethora of hydrogen bonds create 1D chains and 2D layers in the crystal structures of **7**·4MeCN and **8**·2.4MeCN, respectively. ^1^H NMR studies suggest that the crystal structures of the clusters do not persist in DMSO. The two solid complexes emit green light at 20 °C upon excitation at 400 nm; the light emission is ligand-centered.


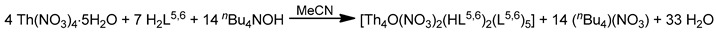
(5)

## 7. Dinuclear and Oligonuclear Uranium-Schiff Base Complexes

For the hypothetical question, “Which is the most controversial element in the Periodic Table —U would be the first candidate because of the unrest over its use and over the consequences of its use [37]. Romans used U minerals as pigments; this use continues today in the preparation of uranium glasses (or “vaseline glasses”) which are impressive for their yellow color and green fluorescence. The existence of the metal was recognized by Klaproth in 1789, but the element itself was actually isolated by Péligot in 1841. It was named after the planet Uranus, which had been discovered by Herschel in 1781. However, it was not until the discovery of U fission by Lise Meitner, Otto Hahn and Fritz Strassman that it became commercially important. Pitchblende (approximating the formula UO_2_) and carnotite, K_2_(UO_2_)_2_(VO_4_)_2_·3H_2_O, are the main uranium minerals, but there are many others. The metal corrodes in air; it is attacked by H_2_O and dilute acids, but not alkalis. It reacts with H_2_, F_2_, Cl_2_ and H_2_O under heating to give UH_3_; UF_6_; a mixture of UCl_4_, UCl_5_ and UCl_6_; and UO_2_, respectively. With O_2_, UO_2_ is produced, but upon heating, U_3_O_8_ forms [34]. The chemistry of U is extremely rich [32,33,34,35,36,37]. In some cases, it behaves like transition metals, but in others more like a lanthanoid [36]. In its compounds, oxidation states from II to VI are well characterized, while relativistic effects lead to involvement of 5f orbitals in its valence shell. Compounds in the oxidation states IV and VI are the most stable (Table 1). Haber recognized the catalytic activity of U in NH_3_ synthesis as long ago as 1909. However, the development of coordination and organometallic chemistry (which began in the 1960s) has led to isolation of complexes that have the ability to activate important small molecules, e.g., N_2_, CO and CO_2_, and to catalyzing a plethora of significant reactions; e.g., Diels-Alder additions, olefin polymerization and hydroamination.

The importance of uranium-Schiff base coordination is discussed in Section 4. It is associated with several aspects of contemporary inorganic synthesis, including the functionalization of {U^VI^O_2_}^2+^; supramolecular chemistry, including the ability of uranyl(VI) complexes to act as anion receptors; homogenous catalysis; theoretical chemistry; and separation technologies related to nuclear fuel processing and nuclear waste management.

### 7.1. Dinuclear Uranyl(VI) Complexes

Most of the to-date structurally characterized dinuclear and oligonuclear uranyl(VI) complexes with Schiff-base ligands and the coordination modes of the latter are listed in Table 3. The structural formulae of the Schiff bases (most presented in their neutral forms) and the abbreviations (used in this review) are illustrated in Figure 2, Figure 3, Figure 13 and Figure 18. The coordination modes of some of these ligands are shown in Figure 19. Most of the complexes have been prepared by reactions of uranyl(VI) sources with preprepared Schiff bases. In some cases, template reactions were used. Some representative examples are discussed below. The U^VI^ centers of the uranyl(VI) cations are most often 7-coordinate with a pentagonal bipyramidal (pbp) geometry. Selected examples are discussed below.

Complexes **9** (**9a**, S = DMF; **9b**, S = DMSO) were prepared by a template procedure involving 2,3-dihydroxybenzaldehyde (8 equiv.), 1,5-diamino-3-azapentane (1 equiv.) and UO_2_(O_2_CMe)_2_·2H_2_O (2 equiv.) in the presence of an external base (LiOH, 4 equiv.), in MeOH, under reflux and subsequent crystallization from DMF/Et_2_O or DMSO/Et_2_O [81]. The anion (HL^7^)^4−^ behaves as a 2.2211111 heptadentate ligand. In these molecules, two uranyl(VI) centers occupy the compartmental sites of the polydentate ligand, the fifth coordination position in the equatorial pentagon being occupied on the “outer” metal ion by a coordinated solvent molecule. Comparison of the structures in the solid state and in solution (studied by NMR techniques) reveals some conformational changes.

The Schiff base H_4_L^8^ was synthesized by the 1:4:6 reaction between 1,4-bis[bis(2-aminoethyl)- aminomethyl]benzene·6HCl, salicylaldehyde and NaOH in refluxing EtOH. Complex [(UO_2_)_2_(L^8^)] (**10**) was prepared by the reaction illustrated in Equation (6). The entire centrosymmetric molecule can be depicted as having a 3-step conformation (Figure 20) [82]. The central benzene ring and the equatorial {O_2_N_3_} pentagonal planes are almost parallel, and the two U^VI^ atoms are displaced 1.20 Å above and below the benzene ring being 12.7 Å apart.


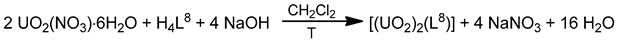
(6)

In complexes [(UO_2_)_2_(H_2_L^9^)_2_(H_2_O)_2_] (**11**) and [(UO_2_)_2_(HL^10^)_2_(H_2_O)_2_] (**12**), the Schiff base ligands are doubly deprotonated [83]. Only one -CH_2_OH group is deprotonated and involved in bridging of the two uranyl(VI) centers.

The 1:1 reaction between the Schiff-base ligand H_3_L^11^, derived from 4,6-*O*-ethylidene-β-d-glucopyranosylamine, and UO_2_(O_2_CMe)_2_·2H_2_O in MeOH gave complex [(UO_2_)_2_(HL^11^)_2_] (**13**) at a 73% yield [84]. The 2-OH group of the saccharide moiety binds in its deprotonated form and bridges the two metal centers giving rise to a four-membered {U^VI^_2_(μ-OR)_2_} rhomb, whereas the 3-OH group remains neutral and is terminally ligated to one uranyl(VI) cation. Both the six-membered rings of the saccharide moiety of the two dianionic ligands adopt a chair conformation.

The reaction of 1,3-bis(salicylideneamino)-2-propanol (H_2_L^13^) and UO_2_(NO_3_)_2_·6H_2_O in the presence of an equivalent of Et_3_N as a base in MeOH-CHCl_3_ under reflux and subsequent crystallization of the resulting powder from DMF gave complex [(UO_2_)_2_(OH)(L^13^)(DMF)_2_] (**15**); Equation (7). The uranyl(VI) cations are bridged by the alkoxido group of the trianionic ligand and the hydroxido oxygen atom (Figure 21) [86]. 


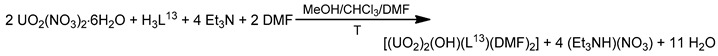
(7)

The reactions of UO_2_(NO_3_)_2_·6H_2_O with the asymmetric {3O,N} Schiff bases H_3_L^14^, H_4_L^15^ and H_3_L^16^ in the presence of base in refluxing EtOH yielded complexes [(UO_2_)_2_(HL^14^)_2_] (**16**), [(UO_2_)_2_(H_2_L^15^)_2_] (**17**) and [(UO_2_)_2_(HL^16^)_2_] (**18**), respectively. All complexes exhibit a symmetric {U^VI^_2_(μ-OR)_2_} core featuring a distorted pbp geometry around each uranyl(VI) center [87]. Compound **14** reacts with 1 equiv. of Et_3_N_,_ in the presence of excess Ag(NO_3_) in DMF, yielding {(CH_3_CH_2_)_3_NH}_2_[(UO_2_)_2_(NO_3_)_2_(sal)_2_] (**16a**), where the geometry around uranyl(VI) center is hexagonal bipyramidal; sal is the salicylate(2-) ligand. Two-phase (H_2_O/CHCl_3_) extraction studies of uranyl(VI) ions from aqueous solutions at different pH conditions employing H_3_L^16^ indicates better efficiency at higher pH (99%, pH 5). Compound H_3_L^16^ was selected for the extraction studies due to its better solubility in CHCl_3_.

In complexes [(UO_2_)_2_(salophen)_2_] (**19**) and [(UO_2_)_2_(salophen)_2_]·0.5CH_2_Cl_2_ (**19a**), the {UO_2_(salophen)} fragments are held together by the coordination of one of the phenoxido oxygen atoms of each salophen^2−^ to the fifth equatorial coordination site of the other {UO_2_(salophen)} moiety [88]. It was demonstrated by UV–Vis spectroscopy that complex **19** retains its dimeric structure in solution of noncoordinating solvents such as CH_2_Cl_2_ and CHCl_3_, while it is equilibrated with [(UO_2_(salophen)(S)] (S = DMF, DMSO) upon addition of S; see Equation (8). The equilibrium constants and formation enthalpy and entropy of the equilibrium between the monomer and the dimer were evaluated from UV–Vis and ^1^H NMR spectral changes. These thermodynamic parameters suggest differences in the coordination abilities of S to {UO_2_(salophen)} (DMF<DMSO) and the solvent effect on the formation of **19** (CH_2_Cl_2_<CHCl_3_).


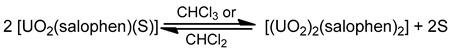
(8)

The Schiff basses H_3_L^13^ and H_3_L^18^ have identical backbones, and complexes [(UO_2_)_2_(OH)(L^13^)- (DMF)_2_] (**15**), Figure 21, and [(UO_2_)_2_(OH)(L^18^)(DMF)_2_] (**22**), have completely similar structures [86,91].

Complex (Et_3_NH)_2_[(UO_2_)_2_(H^HO^salophen)_2_] (**23**) was prepared by the template reaction of 1,2-phenylenediamine (1 equiv.), 2,3-dihydroxybenzaldehyde (2 equiv.) and UO_2_(O_2_CMe)_2_·2H_2_O (1 equiv.) in refluxing MeOH and crystallization from Et_3_N/Me_2_CO [92]; H^HO^salophen is the trianionic ligand derived from triple deprotonation of H_4_^HO^salophen. The dinuclear anion (Figure 22) presents a crown-ether-type coordination site [92], analogously to the more familiar 12-crown-4 motif. This structural feature raised the question of whether such a dimeric species could possess an affinity towards alkali metal cations, analogously to the well-known crown ethers. ESI mass spectrometry experiments have shown that this rigid dimeric species is able to bind the Li^+^ cation with good selectivity over larger Na^+^ and K^+^ ions.

The reaction between [UO_2_(H_2_^Me^L^1^)(THF)] and KH at –78 °C formed a brown solid that upon crystallization from benzene, generated the dimeric hydroxo uranyl(VI) dimer [K_2_(UO_2_)_2_ (OH)_2_(H_2_^Me^L^1^)_2_(C_6_H_6_)_2_] (**24**) at moderate yields (≈35%); H_2_^Me^L^1^ is the dianion of the ditopic Schiff-base pyrrole macrocycle H_4_^R^L^1^ (R = Me) shown in Figure 13. The incorporation of KOH in this complex most probably resulted from the use of impure KH; i.e., some KOH might be present. The ^1^H NMR spectrum of **24** in C_6_D_6_/THF supports the presence of the OH^−^ group with a broad resonance at 10.15 ppm; it also suggests that the Pacman solid-state structure is retained in solution with two separate resonances for the imine protons at 8.75 and 7.89 ppm associated with different N_4_-compartments [93]. Each uranyl(VI) cation is complexed by the four N atoms in one half of the dianionic macrocycle (the half in which the two pyrrolic nitrogens are deprotonated). The pbp coordination environment at each uranyl(VI) group is completed in the equatorial plane by a μ-OH^−^ ligand, which is also coordinated to a K^I^ center. The potassium cations reside above and below the planes of the macrocycle, each at a distance approximately equidistant from the *exo* uranyl(VI) oxygen atoms and the oxygen atom of the hydroxido ligand. The coordination sphere of each K^I^ is completed by an η^2^-interaction with a benzene molecule. Hexavalent uranyl complexes with CCIs interactions to a potassium cation are very rare.

The interesting ligand H_5_L^19^ was synthesized from the 1:2 reaction between 1,3-diamino-2-propanol and 3-formylsalicylic acid in refluxing EtOH. Its reaction with 2 equiv. of UO_2_(O_2_CMe)_2_·2H_2_O, in the presence of Et_3_N (2 equiv.), in refluxing MeOH, gave a solid, the crystallization of which from DMSO afforded complex [(UO_2_)_2_(OH)(H_2_L^19^)(DMSO)_2_] (**25**) [94]. The two uranyl(VI) groups are doubly bridged by the central alkoxido oxygen atom of the 2.211000011 (H_2_L^19^)^3−^ ligand and the hydroxido oxygen atom. Surprisingly, the two carboxylic groups of the trianionic ligand remain neutral in the complex; both form very strong intramolecular H bonds with the neighboring coordinated phenoxido oxygen atoms. 

Expansion of the Schiff-base polypyrrolic macrocycle H_4_^Me^L^1^ allows the formation of an alkali-free uranyl(VI) complex with co-linear uranyl ions and a very short oxido-oxido distance. The reaction of the free anthracenyl macrocycle H_4_^R^L^A^(R = Et) and 2 equiv. of [UO_2_{N(SiMe_3_)_2_}_2_(py)_2_] in py yields a mixture of two products, one of which is [(UO_2_)_2_(^Et^L^A^)(py)_2_] (**26**) [95]. Both uranyl(VI) cations display pbp geometries with the macrocyclic N_4_-donor set and the py N-donor atom comprising the equatorial donor ligands of each metal ion, with the two oxido atoms being mutually *trans* (O=U=O angles of 174.0° and 176.0°). The most notable feature is the short oxido⋯oxido separation between the neighboring uranyl(VI) cations, which is ≈2.71 Å within the molecular cleft. The two N_4_-donor sets remain approximately co-planar, subtending an angle of ≈17° due to the steric demand of the *meso* ethyl groups. A schematic structural representation of the molecule is illustrated in Figure 23. Preliminary cyclic voltammetry (CV) experiments of **26** [0.2 M (*^n^*Bu_4_N)(BF_4_), Fc^+^/Fc) displayed only an irreversible reduction at Epc = −2.46 V. If this is assigned to a reduction of the two non-communicating U^VI^ centers, it represents a difficult reduction.

Exposure of a THF or py of [K_2_(U^V^O_2_)_2_(^Me^L^1^)] (vide infra) to dioxygen results in instantaneous oxidation and the formation of the dinuclear peroxido complex [K_2_(U^VI^O_2_)_2_(O_2_)(^Me^L^1^)] (**27**), Equation (9), isolated at a 55% yield. Oxidation of the same starting uranyl(V) complex with pyridine-N-oxide (pyO) instead of dioxygen in py under an inert atmosphere upon heating gives the mono-oxido-bridged complex [K_2_(U^VI^O_2_)_2_(O)(^Me^L^1^)] (**28**) at a moderate yield [96]; see Equation (10). The solid-state structure of **27** depicts a wedge-shaped, Pacman macrocycle, with symmetrical occupation of each of the N_4_-donor pockets by *trans*-(OU^VI^O)^2+^ groups. The peroxido group is coordinated in a 2.22 manner. The fifth and sixth equatorial donors to each hexagonal bipyramidal uranyl(VI) dication are provided by the two bridging peroxido atoms. The K^I^ centers coordinate to uranyl(VI) oxido atoms and to the peroxido oxygen atoms. The solid-structure of **28** is similar to that of **27**, with occupation of the tetra-anionic Pacman ligand by two uranyl(VI) dications in adjacent N_4_-donor pockets; in contrast to **27**; however, a single oxido group, rather than peroxide, bridges the two metal ions at the obtuse angle U^VI^-O-U^VI^ angle of ≈136°, resulting in pbp uranium geometries. Analogously to **27**, the uranyl(VI) …K^I^ CCIs are maintained. The ^1^H NMR spectrum of **28** displays 14 resonances for the main Pacman skeleton (methyl groups not included), indicating that the asymmetry in the solid-state structures due to K^I^ coordination is retained in solution. 


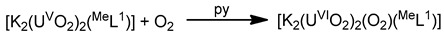
(9)


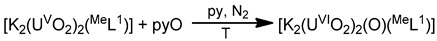
(10)

Reactions of 1 or 2 equiv. of the sodium salt of HL^20^ with [UO_2_Cl_2_(THF)_3_] in THF at room temperature gives the dark red complex [(UO_2_)_2_Cl_2_(L^20^)_2_] (**29**) at a 63% yield [97]. The two uranyl(VI) centers are bridged by the two chlorido groups, while a O,N,S-tridentate chelating (L^20^)^−^ ligand completes the coordination at each metal ion. The coordination of the soft S atom to {UO_2_}^2+^ is extremely rare.

In the dinuclear complex (Me_4_N)[(UO_2_)_2_(OH)(L^21^)_2_] (**30**), the two uranyl dications are singly bridged by the hydroxido group; the (L^21^)^2−^ dianion behaves as a tetradentate chelating ligand (1.110011) with the ether oxygen atoms remaining uncoordinated [98]. The ligand has absolute R,R configuration at the stereogenic carbon atoms of the ethylene bridge. The Me_4_N^+^ cations are sandwiched between the dinuclear anions through weak H bonds and cation…π interactions.

The reaction of the bis-salophen-type ligand *N*,*N*′,*Ν*″,*Ν*″′-tetra-(3,5,-di-tert-butylsalicylidene)- -1,2,4,5-phenylenetetramine Schiff base H_4_L^22^ with UO_2_(NO_3_)_2_·6H_2_O in refluxing Me_2_CO gave the dark red complex [(UO_2_)_2_(L^22^)(Me_2_CO)_2_] (**31**) in high yield. The IR band at 914 cm^−1^ is assigned to the antisymmetric stretching vibration of the *trans*-{O = U^VI^ = O}^2+^ group, *v*_as_(UO_2_). The uranyl(VI) groups are situated in the N_2_O_2_ pockets of the tetra-anionic ligand with a coordinated Me_2_CO molecule completing a pbp geometry at each metal ion (Figure 24) [99]. The CV of the complex in CH_2_Cl_2_ exhibits two quasi-reversible redox processes at E^1^_1/2_ = 0.456 and E^2^_1/2_ = 0.822 V, which are assigned to two sequential one-electron oxidations, where two of the four coordinated phenolato (phenoxido) groups are converted to phenoxyl radicals.

The mononuclear receptor complex [UO_2_(L^23^)] was synthesized in three steps. The first involved nitration (HNO_3_/H_2_SO_4_) of the commercially available benzo-15-crown-5 in CHCl_2_ and the second involved reduction of the dinitro compounds with N_2_H_4_·H_2_O and 10% Pd/C in EtOH. The final step involved template Schiff-base condensation reaction between the diamino compounds, 2 equiv. of salicylaldehyde and 1 equiv. of UO_2_(O_2_CMe)_2_·2H_2_O in EtOH. Comprehensive studies on the solid-sate structures with various alkali and ammonium halides revealed the identification of two cluster motifs. In the first motif exemplified by complex [Li_2_(UO_2_)_2_Cl_2_(L^23^)_2_(H_2_O)_2_] (**32**), there are separated ion pairs, with the Li^I^ cations bound in the crown ether unit, and one chlorido anion coordinated to each uranyl(VI) center. The dimerization is achieved through Li^I^…O = U^VI^ = O interactions. The equatorial plane of each pbp uranyl(VI) center consists of the two imino nitrogen atoms and the two phenolato oxygen atoms of the salen^2−^ unit of (L^23^)^2−^, and one chlorido ligand. An example of the second motif is complex [Na_2_(UO_2_)_2_Br_2_(L^23^)_2_(H_2_O)(MeOH)] (**33**). This complex was prepared by slow diffusion of Et_2_O into a MeOH solution of the receptor [UO_2_(L^23^)] with NaBr added in excess in H_2_O. One Br^−^ ion is weakly coordinated to each Na^I^ center, while the two coordinated solvent molecules are bound to the uranyl dications. The overall configuration is characterized as a stacked packing structure of the receptors with contact ion pair [100].

Heating of the dinuclear complex [(UO_2_)_2_(H_2_L^24^)_2_(DMSO)_2_] (**34**), prepared by a template reaction between 2-hydroxy-3-methoxy-benzaldehyde, tris(hydroxymethyl)aminomethane and UO_2_(NO_3_)_2_- ∙6H_2_O (1:1:1) in refluxing MeOH and subsequent crystallization from DMSO, at 800 °C in an open atmosphere, gives U_3_O_8_ nanoparticles. The nanoparticles are efficient catalysts for the oxidation of alcohols to the corresponding aldehydes using PI(O_2_CMe)_2_ as an oxidant in ethylenedichloride (EDC) [101].

Work from our group [15] has provided access, among others, to compound [(UO_2_)_2_(L^6^)_2_(EtOH)_2_] (**35**). In the dinuclear molecule, the two *trans*-{UO_2_}^2+^ groups are bridged by two phenoxido (phenolato) oxygen atoms that belong to the methyliminophenolate parts of the two doubly deprotonated 2.211 ligands (Figure 14). The ^1^H NMR spectrum of the complex in DMSO-d_6_ provides evidence that the structure is retained in solution with a possible replacement of the coordinated EtOH molecules by solvent molecules.

Complex [Li_2_(UO_2_)_2_Cl_4_(L^25^)(THF)_5_] (**36**) was synthesized by the 1:2 reaction between Li_2_(L^25^) and [UO_2_Cl_2_(THF)_3_] in THF; the yield was ≈45%. It is soluble in THF, but quickly decomposes in the presence of py, CH_2_Cl_2_ and MeCN. It is temperature-sensitive in both solution and the solid state. This temperature sensitivity of **36** is a consequence of a facile (L^25^)^2−^ oxidation reaction. Its solid-state molecular structure reveals two {UO_2_Cl_2_} fragments that are bridged through a highly puckered, bis bidentate N_2_ + N_2_ bridging ligand that behaves in a 2.1111 manner (Figure 25) [102]. The most notable structural feature of the complex is the presence of a [Li(THF)_2_]^+^ cation that coordinates to both of the *endo* uranyl(VI) oxido ligands. The structure also features a [Li(THF)_3_]^+^ cation which is weakly bound to a chlorido ligand. A surprising fact of this reaction is the ligation of two uranyl(VI) fragments to a single (L^25^)^2−^ ligand; another is the absence of LiCl salt elimination. The relatively Lewis-acidic Li^I^ may actually play a template role during the formation of the complex. The ^1^H NMR spectrum of **36** in THF-d_8_ exhibits singlets at 8.11 and 7.22 ppm, which correspond to the two aryl-CH environments. 

Complex (Et_3_NH)[(UO_2_)_2_(O_2_CMe)(L^26^)_2_] (**37**) was prepared through a template reaction involving the dihydrochloride salt of salicylaldehyde S-benzylisothiosemicarbazone, salicylaldehyde and UO_2_(O_2_CMe)_2_∙2H_2_O in hot EtOH; the yield was ≈85%. The two uranyl(VI) dications are bridged by a 2.11 acetate group and each (L^26^)^2−^ ion behaves as a tetradentate chelating O_2_N_2_ ligand (1.111100) [103].

The template reaction of equimolar amounts of 2,6-diformyl-4-methylphenol, *N*-(hydroxyethyl)ethylenediamine and UO_2_(O_2_CMe)_2_∙2H_2_O in refluxing MeOH affords the dinuclear complex [(UO_2_)_2_(HL^27^)_2_] (**38**) at a 65% yield [104]. In the molecule, the U^VI^ centers are in edge-shared pbp N_2_O_5_ coordination spheres, assembled by the two meridional ONNO bridging dianionic ligands which adopt the 2.21011 coordination mode, and two pairs of mutually *trans*-oriented oxido groups. The complex is redox-active and displays two successive U-centered one-electron reductions at E_pc_ = −0.71 and −1.03 V in DMF solution. The complex was used as heterogeneous catalyst for electrochemical H_2_ evolution from aqueous medium at pH 7 with a turnover frequency (TOF) of 384 h^−1^ (the time of the reaction was 5 h); the Faradaic efficiency was 84%. On the other hand, using the **38**-TiO_2_-N719 (dye) composite in photocatalytic H_2_ production in neutral aqueous medium under visible light irradiation, a TOF of 172 h^−1^ with an apparent quantum yield of 7.6% was obtained; the time of the reaction was 4 h. Complex **38** is the first uranyl(VI) complex that can act as both an electrocatalyst and a photocatalyst for the H_2_ evolution reaction in neutral aqueous medium [104].

### 7.2. Trinuclear and Tetranuclear Uranyl(VI) Clusters

The to-date structurally characterized trinuclear and tetranuclear uranyl(VI)-Schiff base clusters and the coordination modes of the ligands have been incorporated in Table 3. The structural formula of H_2_L^28^, whose dianion has been used for the construction of trinuclear clusters, is shown in Figure 18. The H_2_salen ligand, whose dianion has been employed for the synthesis of the only known tetranuclear cluster, is illustrated in Figure 2 (R′ = H, Y = CH_2_CH_2_, X = H). The coordination modes of both ligands are presented in Figure 19.

The monomeric species *trans*-{UO_2_}^2+^ is stable in acidic solutions (pH < 3), but it undergoes a condensation reaction in aqueous solutions at pH>3, as shown in Equation (11). Mononuclear {UO_2_}^2+^, dinuclear {(UO_2_)_2_(OH)_2_}^2+^, and trinuclear {(UO_2_)_3_(OH)_5_}^2+^ and {(UO_2_)_3_(OH)_4_}^2+^ species coexist in aqueous solution at pH = 3–5 [35,105]. Two years ago, the group of Yoshimura performed an excellent study on the proton self-exchange reaction of μ_3_-oxido-/μ_3_-hydroxido-bridged trinuclear uranyl(VI) complexes with the tridentate Schiff base H_2_L^28^.


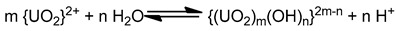
(11)

The new complexes (Et_3_NH)[(UO_2_)_3_(OH)(L^28^)_3_] (**39**) and (Et_3_NH)_2_[(UO_2_)_3_(O)(L^28^)_3_] (**40**) were prepared by the reactions shown in Equations (12) and (13), respectively. A reversible structural conversion between the dianionic and monoanionic trinuclear ions was conducted by protonation/deprotonation of the central μ-O^2−^/μ_3_-OH^−^ coordinated group. The activation enthalpy and entropy of the proton self-exchange reaction between [(UO_2_)_3_(OH)(L^28^)_3_]^−^ and [(UO_2_)_3_(O)(L^28^)_3_]^2−^, determined from temperature-dependent ^1^H NMR studies, are ΔH^ǂ^ = 23 ± 2 kJ∙mol^−1^ and ΔS^ǂ^ = −77 ± 5 J∙K^−1^∙mol^−1^. The IR spectra of **39** and **40** show the *v*_as_(UO_2_) vibrations at 918 and 895 cm^−1^, respectively. Each tridentate 2.211 (L^28^)^2−^ ligand coordinates to three equatorial positions of a pbp uranyl(VI) dication with one phenolato oxygen atom bridging to a neighboring metal center in μ-fashion. The three U^VI^ ions are bridged by a μ_3_-hydroxido (**39**) or a μ_3_-oxido (**40**) group, creating a triangular topology. The μ_3_-O^2−^ ion in **40** is sp^2^ hybridized with an average U^VI^-(μ_3_-O)-U^VI^ angle of ≈118°. The corresponding oxygen atom of the μ_3_-OH^−^ group in **39** is sp^3^ hybridized with an average U^VI^-(μ_3_-O_hydroxido_)-U^VI^ angle of ≈110° [105]. Because of steric hindrance from the *t*-butyl substituents on one phenol ring in (L^28^)^3−^, the phenolato(phenoxido) atom that belongs to the non-substituted aromatic group selectively bridges the uranyl(VI) ions to give the trinuclear anions (Figure 19). 


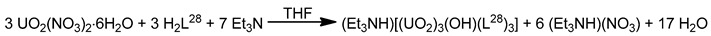
(12)


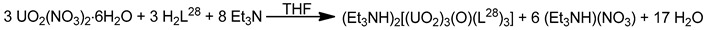
(13)

In an excellent study aiming at the investigation of CCIs in uranyl(V) complexes [106], the group of Mazzanti isolated the interesting decomposition product [Li_4_(UO_2_)_4_(O)_2_(salen)_4_] (**41**), which is the only tetranuclear uranyl(VI) cluster reported to-date. The lower stability of {U^V^O_2_}^+^ complexes of salen^2−^ in the presence of Li^I^ is probably the consequence of both steric and electronic effects associated to the higher charge/ionic radius of Li^+^ (relative to K^+^ and Rb^+^, vide infra). The structure of **41** (Figure 26) consists of four uranyl(VI) groups arranged in a tetrahedral topology. The salen^2−^ groups behave as 3.2211 ligands (Figure 19), each connected to a U^VI^ center through a chelating N_2_+O_2_ mode and to two Li^I^ ions through the deprotonated oxygen atoms. Each Li^I^ ion is quadrupally “bridged” to two salen^2−^ oxygen atoms, one uranyl(VI) oxido atom and one bridging (μ_4_) oxido group; each of the latter bridges two U^VI^ and two Li^I^ centers.

### 7.3. Uranium Complexes at the Oxidation States III, IV and V

Twenty years ago, it was believed that U chemistry was confined to the *trans*-{U^VI^O_2_}^2+^ cation; i.e., the uranyl(VI) dication. This linear species contains covalently bound, axial oxido groups that exhibit practically no chemistry. In the last 10 years or so there has been an explosive growth in the chemistry of this element at lower oxidation states (III, IV, V). Complexes of low-valent uranium (e.g., III and IV) are attracting the intense interest of researchers due to their ability to promote the activation and functionalization of small molecules (N_2_, CO, CO_2_, NO, arenes, alkynes, etc.) under mild conditions. Because of the variable coordination and bonding properties of uranium, its complexes can, in principle, be used in catalytic transformations of small molecules providing an attractive potential to transition metals.

As far as the oxidation state V is concerned, it is now well known that under anaerobic conditions, one-electron reduction of uranyl(VI) complexes can occur, providing access to uranyl(V) compounds that do not disproportionate (see Section 4), although the reactions sometimes proceed further to U(VI) species. Reduction reactions increase the oxido basicity of {U^V^O_2_}^+^, giving rise to oxido-donor interactions to Lewis acidic ions. This makes reduced uranium oxido compounds better models for the heavier, very dangerous (due to high radioactivity) transuranium actinyl cations {AnO_2_)^2+^ (An = Np, Pu; *n* = 1, 2) for which clustering behavior is problematic in the PUREX separation processes for civil nuclear waste treatment. Actinoid oxido bridges also facilitate electron-transfer reactions in environmental waste remediation, generate complexes with interesting magnetic properties and enrich the chemistry of actinoids in minerals. The chemistry of the traditionally inert oxido group of {U^VI^O_2_}^2+^ is also of academic interest, because this lack of creativity is in contrast with the lighter group 6 congeners; e.g., the {Cr^VI^O_2_}^2+^ (chromyl) species.

Polydentate Schiff-base ligands are particularly good supporting groups for the synthesis of uranium complexes with the metal at lower oxidation states or for mixed-valence complexes. Most of the to-date, structurally characterized, dinuclear and oligonuclear U(V), U(VI) and U(III) complexes, the mixed-valence complexes with Schiff-base ligands and the coordination modes of the latter are listed in Table 4 and Table 5. The structural formulae of the Schiff bases (most presented in their neutral forms) and their abbreviations used in this review are illustrated in Figure 2, Figure 3 and Figure 18. The coordination modes of some of the ligands are shown in Figure 19. All synthetic procedures were conducted under anaerobic conditions.

### 7.4. Dinuclear and Oligonuclear Uranyl(V) Complexes

Mazzanti’s group reported the first example of a {U^V^O_2_}^+^ cluster stabilized by CCIs; the cluster is highly stable in organic solvents, and, surprisingly, is stable towards hydrolysis. The ligand of choice was the non-bulky Schiff base salen^2−^ (Y = CH_2_CH_2_ and R′ = X = H in Figure 2) which can stabilize pentavalent uranyl groups through the formation of a very stable cation-cation interaction. The reaction of {[K(U^V^O_2_)I_2_(py)_7_]}_n_ (vide supra) with K_2_(salen) in py and crystallization of the violet powder from pyridine by addition of [18]-crown-6(18C6) and *n*-hexane gave complex [K(18C6)(py)]_2_[K_2_(U^V^O_2_)_4_(salen)_4_] (**42**) in 55% yield; Equation (14). The anion consists of a tetramer of {U^V^O_2_}^+^ groups that forms a square plane. Two potassium ions are located above and below this plane interacting with four uranyl(V) oxygen atoms and four different 3.2211 salen^2−^ oxygen atoms. The U^V^ centers adopt a pbp geometry, by two *trans* oxido groups, two nitrogen atoms, two oxygen atoms from a salen^2−^ ligand and one bridging oxido atom from an adjacent {U^V^O_2_}^+^ group. Complex **42** was also obtained by the reduction of the uranyl(VI) complex [U^VI^O_2_(salen)(py)] in py with [Co^II^(Cp*)_2_] after addition of [KI(18C6)]; Equation (15). Variable-temperature magnetic susceptibility data provide evidence for antiferromagnetic exchange interactions between the 5f^1^ U^V^ centers [107]. The synthesis of **42** was a breakthrough in uranium(V) chemistry providing a tool (the CCIs), previously suggested as a reaction pathway that promotes decomposition, for stable uranyl(V) species.


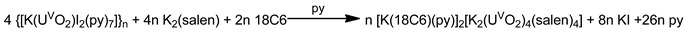
(14)


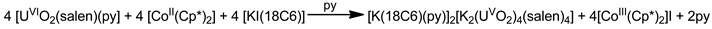
(15)

The ligand salen^2−^ gives the rather analogous complex [Rb_4_(U^V^O_2_)_4_(salen)_4_(18C6)_2_] (**46**) with rubidium(I) [106]. Both **42** and **46** have a highly stability in py with respect to disproportionation. However, some structural differences are observed in **46** as a consequence of the larger size of the Rb^+^ cation. One of the differences is that the [Rb(18C6)]^+^ cations are not isolated, and thus all four Rb^I^ ions can be considered as a part of the tetranuclear uranyl(V) cluster. It should be remembered at this point that Li^I^ can not stabilize uranyl(V) complexes, giving instead the tetranuclear uranyl(VI) cluster [Li_4_(U^VI^O_2_)_4_(O)_2_(salen)_4_] (**41**) (Table 3), which was obtained from a complicated decomposition/disproportionation reaction [106].

The dianionic tetradentate ligands (L^29^)^2−^ and salophen^2−^ can also give clusters containing four uranyl(V) groups and potassium ions. A representative member of this family is complex [(K18C6)(py)]_2_[K_2_(U^V^O_2_)_4_(L^29^)_4_] (**43**), which was prepared by a reaction analogous to that illustrated in Equation (14), replacing K_2_(salen) with K_2_(L^29^) [106]. ^1^H NMR studies in py-d_5_ show that **43** is stable with respect to disproportionation over 30 days. The structure of **43** is similar to that of **42**, suggesting that the coordination properties of (L^29^)^2−^ are very similar to those of salen^2−^ in this type of chemistry. On the contrary, the reaction of the fully aromatic analogue of the salen^2−^ ligand, salophen^2−^, with the uranyl(V) polymer {[K(U^V^O_2_)I_2_(py)_7_]}_n_ in py, has a different outcome. ^1^H NMR studies show that this reaction leads to the formation of a complicated mixture of disproportionation products. When 18C6 was added to the K_2_(salophen) solution—before reacting it with the uranyl(V) starting material and the convenient crystallization from THF—it led to cluster [K(18C6)(THF)]_2_[K_6_(U^V^O_2_)_4_(salophen)_4_I_2_(18C6)_2_]I_2_ (**45**) at a moderate yield. A pyridine solution of **45** is stable towards disproportionation for up to one month in py; ^1^H NMR studies suggest the presence of mononuclear species in solution. The stability of **45** in py is probably associated with its mononuclear form, while, in the absence of the crown ether, the potassium ion leads to the formation of a reactive tetranuclear species. This complex shows a structure similar to that found for the salen^2−^ and (L^29^)^2−^ tetranuclear complexes with four uranyl(V) centers linked with the same T-shaped CCI to form a square form. However, in **45** two additional K^I^ ions are located on opposite sides of the tetranuclear complex, bridging uranyl(V) oxido atoms from two different U^V^ centers. The charges of these two K^I^ ions are balanced by two coordinated I^−^ ions [106]. 

In an attempt to substitute the equatorial chlorido ligand in [U^VI^O_2_Cl(L^30^)] with 2,6-diisopropylanilide (in order to shift the reduction potential of the uranyl group), the groups of Love and Arnold instead obtained the paramagnetic, dinuclear uranyl(V) complex [(U^V^O_2_)_2_(L^30^)_2_] (**52**) in toluene (the yield was ≈65%); (L^30^)^−^ is the monoanion of the dipyrrin ligand HL^30^ [111]. The near-IR electronic spectrum of **52** exhibits a band at ≈6800 cm^−1^ consistent with an 5f–5f transition and U^V^. The reduction reaction presumably proceeds through the formation of a transient {U^VI^O_2_}^2+^/anilide/(L^30^)^−^ species which then undergoes U–N homolysis. The solid-state molecular structure of **52** (Figure 27) reveals a diamond-shaped oxido, bridging between the two uranyl(V) moieties. Complex **52** exhibits rich reactivity; for example, its reaction with 4 equiv. of B(C_6_F)_3_ yields the borane oxido-functionalized compound [U^V^{OB(C_6_F_5_)_3_}_2_(L^30^)] [111]. Cyclic voltammetry studies of this complex in *o*-difluorobenzene suggests that no communication occurs between the two metal ions in the dimer and no ligand-based reduction is observed. A quasi-reversible wave at −0.28 V (F_C_/F_C_^+^) has been assigned as the U^V^/U^VI^ oxidation. 

The exploitation of the Pacman ligand framework, e.g., ligands H_4_^R^L^1^ (Figure 13) and H_4_^R^L^A^ (Figure 18), has led to the discovery of a variety of new reaction schemes that enable the reductive functionalization of the {U^VI^O_2_}^2+^ group in a controlled manner. Representative examples are described below.

The reaction between the mono(uranyl) Pacman complexes [U^VI^O_2_(H_2_^Me^L^1^)(S)] (S = THF, py) and 1.5 equiv. of the uranyl(VI) silylamide [U^VI^O_2_{N(SiMe_3_)_2_}_2_(py)_2_] in py at 120 °C for 12 h gives the paramagnetic complex [(Me_3_SiOU^V^O)_2_(^Me^L^1^)] (**48**) in 25% yield [108]. A similar reaction between the macrocycle H_4_^Me^L^1^ and 2.5 equiv. of [U^VI^O_2_{N(SiMe_3_)_2_}_2_(py)_2_] also yields **48** (37%). The ^29^Si NMR spectrum of the complex exhibits a resonance at 160 ppm assigned to the SiMe_3_ group of the newly silylated oxo group. The IR spectrum shows two bands at 862 and 802 cm^−1^ assigned to U^V^–O stretching vibrations that are weakened and desymmetrized with respect to [U^V^O_2_(H_2_^Me^L^1^)(S)]; the Si–O stretching vibration appears at ≈1100 cm^−1^. A single-crystal X-ray diffraction study of the complex revealed (Figure 28) that the normally robust *trans*-uranyl geometry had been lost following the formation of the {U^V^_2_(μ-O_uranyl_)_2_} core. The complex represented the first case in which the same yl-derived-oxido group was shared by two uranium centers in a mutually *trans* geometry, as a result of the migration of one oxido group to a mutually *cis* position. Another exciting structural feature is that both *exo*-oxido groups are silylated, and the resulting dimer has a short U^V^⋯U^V^ distance of ≈3.36 Å. Each U^V^ center has a pbp geometry with axial oxido/siloxide groups and one N_4_-donor set of the tetra-anionic macrocycle (^Me^L^1^)^4−^ equatorial; the fifth equatorial position, normally occupied by a donor solvent, is occupied instead by an oxido group which was originally bound at an axial position of the adjacent uranyl group. Variable-temperature magnetic susceptibility studies reveal a rather strong antiferromagnetic coupling with a Néel temperature at 17 K. Upon exposure to air for 48 h, a wet benzene solution of **48** shows no change in its ^1^H NMR spectrum, indicating that the complex is stable towards redox decomposition. Concerning the mechanism responsible for the formation of this remarkable compound, it has been suggested that two competing reductive processes occur that involve either U–N or N–Si bond homolysis [108].

Reactions between solution of [(Me_2_AlOU^V^O)(H_2_^Me^L^1^)(py)_2_] (this compound represents the first example of reductive functionalization of the uranyl oxido group by Al^III^) with either 1 or 2 equiv. of the strong base MeLi gives solely [Li_2_(U^V^O_2_)_2_(H_2_^Me^L^1^)_2_(py)_2_] (**51**) in moderate yield (≈40%); Equation (16). Each Li^I^ ion is coordinated to two imine nitrogen atoms; it is also bound to the uranyl *endo*-oxido atom and a py molecule. Each U^V^ center has migrated from its usual N_4_ donor pocket to an alternative pyrrole-imine-imine-pyrrole set; this results in the doubly deprotonated macrocycle folding at the *meso* carbon atoms and not at the aryl groups, and the resulting conformation can be described as “bowl-shaped” geometry [110].


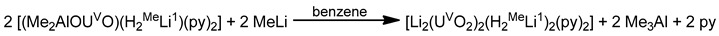
(16)

The highest nuclearity uranyl(V) cluster with Schiff-base ligands are [M^I^_6_(U^V^O_2_)_6_(H_2_^Me^L^1^)_6_(py)_6_] (M = Rb, **53**; M = Cs, **54**). The two clusters were prepared by the reactions which are illustrated in general Equation (17). In contrast to their lighter homologues with alkali cations, which are mononuclear or dinuclear, complexes **53** and **54** crystallize as a ring of the six uranyl(V) Pacman units, linked by the Rb^I^ or Cs^I^ cation through metalation to the *exo* uranyl(V) oxygen atoms of one uranyl(V) group to the *endo* oxygen atom of an adjacent uranyl(V) group [112]. The coordination motifs in the hexagonal assemblies resemble the shape of a crown, similar to 18C6. The cavities of these “uranium crown” molecules are large and empty; in the case of **54**, this cavity has a volume of ≈650 Å^3^, significantly larger that organic molecules such as C_60_ with a diameter of 7 Å.


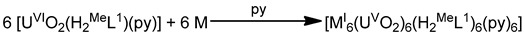
(17)

### 7.5. Dinuclear and Oligonuclear Uranium(IV) Clusters

Several polydentate Schiff bases have been used for the synthesis of dinuclear and oligonuclear (or polynuclear) U(IV) complexes. In many preparative protocols, the starting materials were convenient U(IV) compounds.

A French team in CEA Saclay, France, led by Ephritikhine, Thuéry and Salmon contributed much to the chemistry of U(IV) clusters. Selected examples are described below. Clusters (pyH)_3_[U^IV^_3_(O)Cl_9_(L^31^)] (**55)** and [U^IV^_4_(O)(L^32^)_2_(H_2_L^32^)_2_(py)_2_](CF_3_SO_3_)_2_ (**56**) were prepared using the ligands H_4_L^31^ and H_4_L^32^ (with R = Me), respectively; Equations (18) and (19). Adventitious traces of O_2_ were responsible for the U(III) →U(IV) oxidation during the preparation of **56**. The pbp U^IV^ centers in the triangular trianion of **56** are held together by one μ_3_-oxido group, while all the chlorido ligands are terminal. The tetra-anionic Schiff base bridges the three metal ions in a 3.221111 mode (Figure 19) [113]. The four U^IV^ centers in the dication of **56** are in a distorted tetrahedral topology bridged by the μ_4_-oxido group. The two crystallographically independent metal ions are in different environments; one is 8-coordinate with a dodecahedral geometry and the other is 9-coordinate in a capped square antiprismatic coordination environment. The U^IV^ ions are held together through two 3.221111 (L^32^)^4−^ and two 2.211100 (H_2_L^32^)^2−^ ligands [113]. The (L^32^)^4−^ Schiff-base ligand in the tetranuclear cation does not match exactly the usual boat (umbrella) or stepped conformations, while the (H_2_L^32^)^2−^ ligand is a borderline case of the umbrella shape in which the two aromatic rings of each dianion are facing each other (with a dihedral angle of ≈25°).


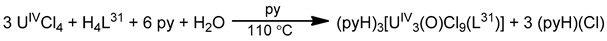
(18)


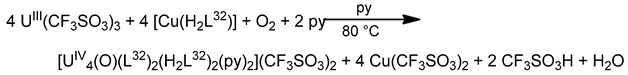
(19)

The use of the potentially hexadentate Schiff base H_4_^HO^salophen (the structural formula of its dianion has been presented in Figure 3) in uranium(IV) chemistry has provided access to the structurally impressive octanuclear cluster (pyH)_2_[U^VI^_8_(O)_4_Cl_10_(^HO^salophen)_4_] (**57**); Equation (20). Each tetra-anionic ligand bridges four U^IV^ centers in a 4.222211 ligation mode [114]. The anionic cluster exhibits a pseudo D_2d_ symmetry and can be viewed as four {U^IV^_2_(O)Cl_2_(^HO^salophen)} dinuclear subunits, which are held together by eight phenoxido bridges from the (^HO^salophen)^4−^ ligands and two μ-chlorido groups. The whole core appears to be {U^IV^_8_(μ_4_-O)_4_(μ-Cl)_2_(μ-OR)_16_}. The octanuclear anion itself is roughly cylinder-shaped, with a height of ≈9 Å and a diameter of ≈18 Å. Two of the four crystallographically independent U^IV^ centers are 8-coordinate with a square antiprismatic geometry, and two are 9-coordinate with a distorted tricapped trigonal prismatic geometry.


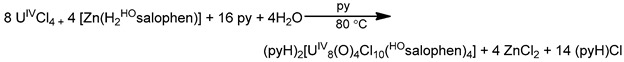
(20)

In addition to **55**, the use of H_4_L^31^ has also yielded the hexanuclear cluster (pyH)_2_[U^IV^_6_Cl_10_(L^31^)_4_(py)_4_] (**63**); see Equation (21). Two of the ligands adopt the 3.222111 coordination mode, while the other two are found in the 5.222111 ligation mode. All the chlorido groups are terminal. Two, out of the three crystallographically independent U^IV^ centers, are 8-coordinate with square antiprismatic and dodecahedral geometries; the third metal ion is 9-coordinate with a tricapped prismatic geometry. The six U^IV^ ions have a zig-zag topology [116]. The crystal structure of clusters **59**–**63** [115,116] underline the remarkable associating capacity of the various anionic forms of the H_4_L^31^, H_4_L^32^, H_4_L^33^ and H_4_L^34^ Schiff bases, by achieving a large number of coordination bonds to U^IV^ centers; this is particularly notable in the hexanuclear complex **63** where two (L^31^)^4−^ ligands are each bound to five metal ions.


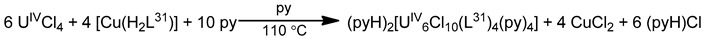
(21)

An interesting, potentially octadentate, tricompartmental macrocyclic Schiff-base ligand is H_4_L^35^. This can be synthesized by the Schiff-base condensation of 2,2′-dihydroxy-3,3′-diformyl-5,5′- di-*tert*-butylbiphenyl and *o*-phenylenediamine; using B(OH)_3_ as a template reagent; the reaction can be carried out at room temperature affording, after crystallization from a mixture of CH_2_Cl_2_ and MeOH, forming yellow prisms of the free ligand in 66% yield. This complex readily gives dinuclear U(IV) complexes (Table 4). In complexes [U^IV^_2_Cl_4_(L^35^)] (**64**) and [U^IV^_2_Cl_4_(L^35^)(py)_2_] (**65**), each metal ion occupies a {N_2_O_2_} cavity of the 2.11111111 tetra-anionic ligand with either a pbp environment in **64**, two chlorido groups (one terminal, one μ) being at the apical positions of each U^IV^, or a dodecahedral environment in **65**, where there is one extra coordinated py molecule at each metal ion [117]. Changing U^IV^Cl_4_ for U^IV^(acac)_4_ in reactions with H_4_L^35^ in THF gave light orange [U^IV^_2_(acac)_4_(L^35^)] (**66**) in 70% yield. In contrast to **64** and **65**, the U^IV^ centers are linked to one {N_2_O_2_} and to the central {O_4_} cavities, and the second {N_2_O_2_} site remains “free” (2.22111100 behavior), Figure 29. This difference might reflect the harder (HSAB) character of the “U^IV^(acac)_2_” fragment, by comparison with the “U^IV^Cl_2_” fragment, leading to the preferential bonding of the former to the harder deprotonated oxygen atoms of (L^35^)^4−^. The coordination sphere of each U^IV^ ion in **66** is completed by the four oxygen atoms of two chelating (1.11) acetyacetonato ligands. The coordination geometry of the two metal ions is square antiprismatic, with the {N_2_O_2_} site defining one square face for one U^IV^ and the {O_4_} site defining one square face for the second U^IV^; the other square face for both metal ions is defined by the four oxygen atoms of two acac^−^ ligands [117]. The variable-temperature magnetic study of **64** and **66** shows the depopulation of the Stark sublevels of U(IV) and a possible antiferromagnetic exchange interaction between the 5f^2^ metal ions.

Treatment of U^IV^(acac)_4_ with H_4_L^35^ in py affords, after 5 days at 80 °C, the light orange complex [U^IV^_2_(acac)_3_(HL^36^)] (**67**) at a 35% yield [117]. The original ligand is, thus, converted into the mono(benzimidazolyl) derivative H_4_L^36^, whose trianion bridges two U^IV^ centers in a 2.11101110 fashion. This transformation is obviously U^IV^-mediated/assisted, since H_4_L^35^ is inert in refluxing py. One U^IV^ ion is linked to the {N_2_O_2_} cavity of (HL^36^)^3−^ and to the oxygen atoms of two chelating (1.11) acetylacetonato ligands; this metal ion is 8-coordinate with a distorted square antiprismatic geometry, and one square face of the polyhedron being occupied by the atoms of the {N_2_O_2_} donor site and the other by the four acac^−^ oxygen atoms. The other U^IV^ ion of the dinuclear molecule also adopts a distorted square antiprismatic geometry, being coordinated to six acac^−^ oxygen atoms—one phenoxido oxygen atom of (HL^36^)^3−^ and the pyridine-type nitrogen atom of the benzimidazolyl moiety of the ligand. The molecular structure of the complex is illustrated in Figure 30.

Complexes **64**–**67** demonstrate the potential of the “calixsalophen”-type macrocycle H_4_L^35^ as a bridging ligand for the design of dinuclear U(IV) complexes. Depending on the nature of the auxiliary ligand, the U^IV^ ions can occupy the {N_2_O_2_} or {O_4_} sites of the compartmental ligand, a property that is potentially useful for the controlled synthesis of heterodinuclear 3d/5f-metal complexes.

Complexes [U^IV^_2_(*cyclo*-salophen)(py)_4_] (**68**) [118] and [U^IV^_2_(L^38^)_2_] (**69**) [119] do not contain Schiff-base ligands; however, they have been prepared from interesting reactions involving Schiff bases. The structural formula of *cyclo*-salophen^8−^ is illustrated in Figure 3 (Section 2). Complex **68** was synthesized from the reaction of [U^III^I_3_(THF)_4_] and K_2_(salophen) in THF/py. The molecular structure of the complex shows the presence of a U(IV) dimer complexed by the octadentate octa-anionic amidophenolate macrocyclic ligand *cyclo*-salophen^8−^ formed from the reductive C–C coupling of the two imine groups of two salophen^2−^ ligands. The synthesis of the complex is accompanied by the formation of the U(IV) complex [U^IV^I_2_(salophen)(THF)_2_]. The formation of **68** probably proceeds through the stepwise or simultaneous reduction of four imine groups by four U^III^ ions to yield unstable U(IV) complexes of the corresponding radical ions, which rapidly couple to form two C–C bonds. In the dimer, *cyclo*-salophen^8−^ behaves as a 2.11112222 ligand, the bridging atoms being the four deprotonated amido nitrogen atoms. The reaction of **68** with 4 equiv. of Ag(CF_3_SO_3_) leads to the cleavage of the two C–C bonds and the disruption of the dimeric structure, affording the mononuclear U(IV) complex [U^IV^(O_3_SCF_3_)_2_(salophen)(THF)] in which the two imine groups of the salophen^2−^ ligand are restored. Thus, the tetradentate Schiff base H_2_(salophen) can be used to stabilize reduced U complexes by storing electrons in C–C bonds formed by reductive coupling of the imine groups. The stored electrons can become available to oxidizing agents through cleavage of the C–C bonds [118].

Complex [U^IV^_2_(L^28^)_2_] (**69**) was synthesized by the reduction of the bis-ligand complexes [U^IV^X_2_(L^37^)_2_] in THF (X = Cl, I) with K metal; HL^37^ (Figure 18) is a tridentate, redox-active Schiff-base ligand [119]. The hexadentate ligand (L^38^)^4−^ is formed through the intramolecular reductive coupling of the imine groups of the (L^37^)^−^ units. The solid-state structure of **69** shows that the dinuclear molecule consists of two {U^IV^(L^38^)} units bridged by the phenolato oxygen atoms of the two 2.211111 ligands. Each U^IV^ center is 7-coordinate in a capped trigonal prismatic coordination environment. The complex is stable for several weeks at room temperature in the solid state or in toluene solution under Ar. Reactivity studies with O_2_ and 9,10-phenanthrenequinone show that **69** can act as a multielectron reducing agent, releasing two electrons through the cleavage of the C–C bond to restore the original imine functional group of the ligand. Electrochemical studies of **69** in py show the presence of irreversible ligand-centered reduction and of a reversible U(IV)/U(III) couple [119].

The bis-^H^salophen^6−^ Schiff-base ligand (R = R′ = H in Figure 3) has also been studied for the synthesis of dinuclear U(IV) complexes [120]. The addition of 1 equiv. of [U^IV^I_4_(OEt)_2_] onto a THF solution of {[Na_2_U^IV^(bis-^H^salophen)]}_n_ and recrystallization from a py/hexane solution gives complex [U^IV^_2_(bis-^H^salophen)(py)_6_]I_2_ (**70**); Equation (22). An analogous reaction with U^IV^Cl_4_ as starting material in THF yields [U^IV^_2_Cl_2_(bis-^H^salophen)(THF)_2_] (**71**); Equation (23). In both complexes, two U^IV^ centers are encapsulated by the octadentate hexa-anionic 2.11112211 ligand, which adopts a helical conformation (Figure 31). Both metal ions in **70** are 8-coordinate with two phenolato, two amido and one imine donors from the bis-^H^salophen^6−^ ligand and three py molecules in a dodecahedral fashion. The two amido units of the ligand act as bridging moieties. The molecular structure of **71** is very similar to the cation of **70**. Due to the coordination of the chloride ions, **71** is neutral, whereas **70** is cationic. There is only one coordinated THF molecule per U^IV^ in **71** and the metal ions are thus 7-coordinate. The two electrons stored in the C–C bond between the two amido moieties in **70** can become available for the reduction of oxidation substrates. For example, the reaction of this complex with 1 equiv. of I_2_ in py has as a result the cleavage of the C–C bond, restoration of the original Schiff-base motif and isolation of [U^IV^I_2_(salophen)(py)_2_)] [120].


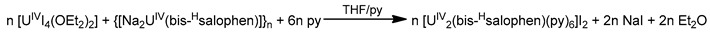
(22)


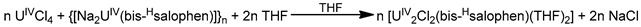
(23)

The Pacman macrocycles (H_4_^R^L^1^, Figure 13; H_4_^R^L^A^, Figure 18) have also been used for the synthesis of dinuclear U(IV) complexes. Complexes [U^IV^_2_(OAr)_2_(S_2_)(^Et^L^A^)] (**72**) and [U^IV^_2_(OAr)_2_(S)(^Et^L^A^)] (**73**) have been obtained by reactivity studies of the soluble complex [KU^III^_2_ (OAr)_2_(BH_4_)(^Et^L^A^)(THF)_2_] (**81**), vide infra; the alkoxide OAr^−^ is OC_6_H_2_*^t^*Bu_3_-2,4,6 [121]. Addition of an excess of S_8_ to a slurry of **81** in toluene resulted in the isolation of the thermally stable product **72** in 41% yield; Equation (24). In the solid-state structure (Figure 32), the intermetallic cleft is occupied by a bridging 2.22 persulfido ion, indicating that the U centers have been oxidized to U(IV). Addition of an excess of CS_2_ to a suspension of **81** in toluene-d_8_ results in a slow color change from dark to brownish orange over a time of 10 min and precipitation of orange **73**. The solution species were characterized by NMR spectroscopy to be [U^IV^_2_(OAr)_2_(CS_2_)(^Et^L^A^)] and **73**. The formation of the latter, which contains a bridging-sulfido group (μ-S^2−^), has been attributed to the slow reductive cleavage of the bound CS_2_ molecule in the former to form S^2−^ and release the unstable CS. This is very unusual transformation since CS is not stable, and thus not prone to eliminate, in contrast to reactions of CO_2_ with reducing metal complexes that can eliminate CO and form oxido-bridged species. The U^IV^ center in **72** and **73** are 7 and 6-coordinate, respectively.


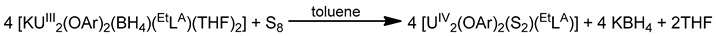
(24)

The group of Love and Arnold managed to prepare a new type of double uranium oxo cation {O-U^IV^-O-U^IV^-O}^4+^ by selective oxygen-atom abstraction from macrocyclic Pacman uranyl complexes using either boranes or silanes [122]. A significant degree of multiple U = O bonding is evident throughout the {U^IV^_2_O_3_} core, but either *trans*-, *cis*- or *trans*-, *trans*-{OUOUO} motifs can be isolated as boron- or silicon-capped oxido complexes. The 1:2 reaction of the dinuclear(VI) complex [(U^VI^O_2_)_2_(^Et^L^A^)(py)_2_] (**26**) and the diborane B_2_(pin)_2_ (pin = pinacolate) in py at 80 °C yields the paramagnetic complex [{pinBO)U^IV^OU^IV^(OBpin)}(^Et^L^A^)(py)_2_] (**74**). Both {U^VI^O_2_}^2+^ groups of **26** have undergone U^VI^→U^IV^ reductions and borylation, and a single oxido-atom abstraction, resulting in the formation of the O(Bpin)_2_ byproduct. This is the first example of the use of diboranes to deoxygenate metal complexes; Equation (25). Complex **26** also reacts with the diborane B_2_cat_2_ (cat = catecholate) in py at 80 °C, leading to the isolation of the cathecholboroxy-analogue of **74** [{(py)catBO}U^IV^OU^IV^(OBcat)- (^Et^L^A^)(py)_2_] (**75**). Complex **75** transforms slowly into the new catecholato(-2)-bridged complex [{U^IV^OU^IV^(O_2_C_6_H_4_)}(^Et^L^A^)(py)_2_] (**76**), which is the product of both boroxy ligands and the addition of a catecholato(-2) ligand, {C_6_H_4_O_2_}^2−^, that bridges the two U^IV^ centers. It was envisaged that **26** could react with other p-block reactants aside from diboranes to give strong new O–E bonds (E = p-block element). Its reaction with the silane Ph_2_SiH_2_ in py at 125 °C yields [{(HPh_2_SiO)U^IV^O- U^IV^(OSiPh_2_H)}(^Et^L^A^)(py)_2_] (**77**); Equation (26).


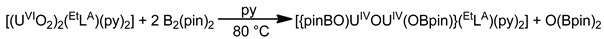
(25)


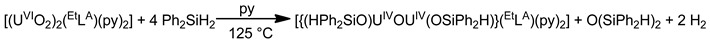
(26)

The molecular structures of **74**, **76** and **77** are illustrated in Figure 33. It comprises two exogeneous boroxide ligands and shows that one *endo* oxido atom has been eliminated with the remaining forming a fused {U^IV^-O-U^IV^} core that is essentially linear. Of great interest is that one reduced {UO_2_}^2+^ group retains the *trans*-{(pinB)OU^IV^O} geometry, but the other has rearranged to a *cis*-{OU^IV^O} configuration. The core of **75** is similar to **74**, possessing axial and equatorial boroxides. However, the catBO^−^ ligand that is axially coordinated to one U^IV^ center in **75** contains an additional py donor molecule. Complex **77** is a siloxy-analogue of **74** and **75** and is only formed in the presence of a catalytic amount (25%) of an alkali metal salt, e.g., K{N(SiMe_3_)_2_} or KO*^t^*Bu, suggesting that a hypervalent silicate facilitates bond homolysis. The reductive deoxygenation of **26** by the diborane is a new reaction type, whose mechanism would involve reaction at the most accessible *exo*-oxido ligand, with B–B bond homolysis forming U^V^-OBpin or U^V^-OBcat, and releasing Bpin/Bcat which can either abstract H atoms from solvent, or react with the other uranyl *exo*-oxido atom. This will result in a reduced U(V) intermediate [(pinBO)U^V^(O)_2_U^V^(OBpin)]^4+^ ion with elongated U^V^ = O*_endo_* bonds that now have greater oxido-basicity, facilitating the electron transfer required from one *endo*-oxido atom to form a covalent *μ*-oxido bridge between the two U^IV^ centers.

### 7.6. Dinuclear Uranium(III) Complexes

The number of uranium(III) complexes supported by Schiff-base ligands is very limited; all complexes are dinuclear. The Pacman Schiff-base macrocycles have played a crucial role in the development of this chemistry. A valuable starting material for U(III) chemistry is the [U^III^(BH_4_)_3_(THF)_2_] synthon, prepared in high yields through the reaction illustrated in Equation (27) [123].


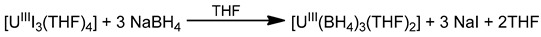
(27)

The reactions between 2 equiv. of [U^III^(BH_4_)_3_(THF)_2_] and alkali metal salts of the Pacman macrocycles H_4_^Me^L^1^ (Figure 13, R = Me), and H_4_^Et^L^A^ (Figure 18, R = Et) and crystallization from THF or py/hexane yield complexes [Li(THF)_4_][U^III^_2_(BH_4_)_3_(^Me^L^1^)] (**78**) and [NaU^III^_2_(BH_4_)_3_(^Et^L^A^)(py)_6_] (**79**), respectively [123]. The ^11^B NMR spectra in THF-d_8_ show two different boron environments in an 1:2 ratio. For example, the spectrum of **78** displays broad resonances at 325 ppm (μ-BH_4_^−^) and 212 ppm (terminal BH_4_^−^).

In the solid state (Figure 34), the anion of **78** does not adopt the classic Pacman geometry, where the *ortho* phenylene rings of the macrocycle act as hinges, but instead the ligand flexes at the *meso* carbon atoms, adopting a bowl-shaped configuration. Each U^III^ center is pseudo-octahedral and bound in the equatorial plane to two *ortho* imine nitrogen atoms of one aryl ring and to two adjacent pyrrolido nitrogens, while two BH_4_^−^ groups are at the axial positions. The anthracenyl macrocycle in **79** adopts the classic Pacman configuration (Figure 35), since the separation between the two U^III^ binding pockets is greater than for the *ortho* phenylene macrocycles [123]. Each metal ion is 7-coordinate with a pbp geometry. Four of the equatorial sites are provided by two imine nitrogen atoms (attached to different anthracene rings) and two pyrrolido nitrogen atoms of the ligand, while the fifth equatorial donor site is occupied by a py molecule. One terminal and one bridging BH_4_^−^ occupy the axial sites. The Na^I^ cation is coordinated to one H atom of a terminal BH_4_^−^ and to four py molecules.

Complexes **78** and **79** combine the reducing potentials of two U^III^ centers with the versatility of the borohydride anion; thus, this chemistry presents excellent opportunities to investigate the reactivity of low oxidation-state uranium confined within a variety of macrocyclic frameworks.

The BH_4_^−^/(^Et^L^A^)^4−^ “blend” has also been used to stabilize the dinuclear complexes [MU^III^_2_(OAr)_2_(BH_4_)(^Et^L^A^)(THF)_2_] (M = Na, **80**; M = K, **81**), where Oar = OC_6_H_2_*^t^*Bu_3_-2,4,6. These complexes can be prepared by substitution of the terminal BH_4_^−^ ligands [those bound to U^III^ outside the macrocyclic cleft of (^Et^L^A^)^4−^] of **79** and its K^I^-analogue (not shown in Table 4) by aryloxido ligands [121]. The reactions result in a single, weakly bound, encapsulated *endo* Na^I^ or K^I^ borohydride bridging the two U^III^ centers. The main difference between the structures of **80** and **81** is the binding of K^I^ and Na^I^ ions within the cleft. The K^I^ center is sandwiched symmetrically between all four pyrrolide rings with K^I^-(pyr)centroid separation of ≈3.15 Å. By contrast, the smaller Na^I^ ion is disordered over two sites about the crystallographic C_2_ axis of the molecule, because it cannot effectively bridge all four pyrrolides. The reactivity of **81** for the activation of small molecules was described in Section 7.5.

### 7.7. Dinuclear and Oligonuclear Mixed-Valence Uranium Complexes

There are few dinuclear and oligonuclear mixed-valence uranium complexes [64,106,107,124]. The ligands salen^2−^, (L^39^)^2−^, *^t^*^Bu^salophen^2−^ and (H_2_^R^L^1^)^2−^ are protagonists in this chemistry. After careful consideration of the electrochemical properties of **42**, the reaction of 0.75 equiv. of [K(U^V^O_2_)(salen)(py)] with 1 equiv. of [U^VI^O_2_(salen)(py)] in py allowed the selective synthesis of [K_3_(U^VI^O_2_)(U^V^O_2_)_3_(salen)_4_(18C6)] (**82**); Equation (28). Complex **82** can also be obtained by chemical oxidation of **42** with CuI; Equation (29). In turn, the reduction of **82** with 1 equiv. of [Co^III^(Cp*)_2_] yields **42**, thereby indicating that the oxidation process is reversible. Similar to the structure of **42**, the molecular structure of **82** is composed of a tetrameric unit that consists of uranyl groups coordinated to each other forming a square plane capped by two “bridging” K^I^ ions. The isolation of **82** demonstrates that the CCIs can provide a strategy to the rational synthesis of mixed-valence actinoid complexes [107].


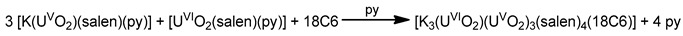
(28)


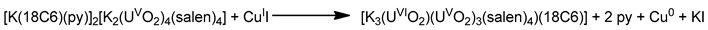
(29)

In line with the above results, the 1:1 reaction between [U^VI^O_2_(salen)(py)] and [Co^III^(Cp*)_2_][U^V^O_2_(salen)(py)] in py leads to the isolation of [Co^III^(Cp*)_2_][(U^VI^O_2_)(U^V^O_2_)(salen)_2_(py)] (**83**), Equation (30), in 92% yield. The dark green dinuclear complex is stable in py for almost one month [106]. The valence is localized in the dinuclear anion (Figure 36). Both U ions are 7-coordinate with a pbp geometry by the four donor atoms of one tetradentate chelating (1.1111) salen^2−^ ligand situated in the equatorial plane and the two uranyl oxygen atoms at the axial positions; the seventh coordination site is occupied by a py nitrogen atom at U^V^ and an uranyl oxygen atom from the {U^V^O_2_}^+^ group at U^VI^.


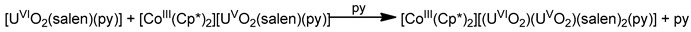
(30)

Complex [(U^V^O_2_)_2_U^IV^_2_(O)(L^39^)_4_] (**84**) was prepared in py by the reaction represented by Equation (31). The structure consists of two {U^V^O_2_(L^39^)⋯U^IV^(L^39^)}^+^ fragments bridged by an oxido (O^2−^) group in a linear manner. In each fragment, the oxido atom of the {U^V^O_2_}^+^ group binds one U^IV^ center. The four uranium and five oxido atoms are coplanar, the metal centers being almost linear (Figure 37). The U ions are 7-coordinate with a pbp geometry [124].


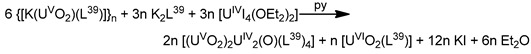
(31)

The reaction of [K(U^V^O_2_)(*^t^*^Bu^salophen)], containing a bulky ligand which was expected to prevent CCIs, with the U(IV) complex [U^IV^Cl_2_(salen)(THF)_2_] resulted in a complicated mixture of products including U(IV), {U^V^O_2_}^+^ and {U^VI^O_2_}^2+^ species [124]. One of the products is [(U^V^O_2_)_2_U^IV^_3_(O)_2_(*^t^*^Bu^salophen)_2_(salen)_3_] (**85**). The structure of **85** consists of five U centers connected to each other by five bridging μ-oxido groups in a quasi planar arrangement. An additional, a triply bridging O^2−^ group is also found at the center of the distorted pentagon connecting one U^IV^ and two U^V^ ions.

The Pacman Schiff-base macrocycles also form mixed-valence U(IV)/U(V) complexes [64]. The reactions between THF solutions of [U^VI^O_2_(H_2_^R^L^1^)(THF)] and [U^III^(Cp)_3_] give the dark orange complexes [(U^V^O_2_)U^IV^(Cp)_3_(H_2_^R^L^1^)(THF)] (H_2_^Et^L^1^, **85a**; H_2_^Me^L^1^, **85b**) in which the dianionic ligands behave in a 2.1111000 manner. The complexes are exclusively *exo*-oxido metalated. In the ^1^H NMR spectra, the two ligand pyrrole NH protons in the *endo* cavity resonate at ≈51 ppm. These highly paramagnetically shifted resonances, along with other macrocyclic ligand resonances, suggest that the {U^III^(Cp)_3_} fragment has reduced the uranyl(VI) to uranyl(V) forming mixed-valence complexes, Equation (32). The uranyl(V) group remains linear (O*_exo_* = U^V^ = O*_endo_* ≅ 177°) and the bimetallic bridge is also almost linear (U^IV^-O*_exo_* = U^V^ ≅ 170°), Figure 38. Complex **85b** behaves as a single-molecule-magnet (SMM) below 4 K, with an effective energy barrier to magnetization reversal of ≈27 K. This value is very similar to previously reported U(V) single-ion magnets (SIMs). This, together with the fact that U(IV) is often non-magnetic in low-symmetry geometries, and with the non-appearance of any clear sign of exchange magnetic interactions between the metal sites in the susceptibility curves, leads to the conclusion that the slow magnetic relaxation is due to the uranyl(V) group.


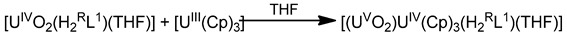
(32)

## 8. Concluding Comments in Brief and Outlook

We hope that this review has provided the readers with a taste of the synthetic chemistry, reactivity studies, structures and some properties of actinoid (Np, Th, U) complexes with Schiff-base ligands. Empirical synthetic routes and strategies have been discussed and critically analyzed through selected examples. Particular attention has been paid to the synthetic rationale behind the reactions and to the criteria for selection of the metal-containing starting materials and the ligands. Most routes and strategies refer to U chemistry, because the number of the Np and Th complexes with Schiff bases as ligands is very limited. 

Ligand design has an important effect on the nature of the products; e.g., on the reductive functionalization of the uranyl(VI) ion. The use of a Pacman-shaped ligand that possesses a phenylene hinge between the top and bottom N_4_-donor pockets enables access to one-electron U(VI)→U(V) reductive functionalization, whereas the two-electron U(VI)→U(IV) reduction is not observed. Alternatively, a Pacman ligand that has an anthracenyl hinge can incorporate two uranyl(VI) moieties, resulting in U(VI)→U(IV) reductive functionalization.

We expect that the synthesis and study of new oligonuclear An-Schiff base complexes will be intensified in the future. Research directions might include: (1) The synthesis and characterization of Pu complexes with Schiff-base ligands. Of course, the great problem here is the fact that only a limited number of research laboratories have the appropriate equipment for the study of transuranium elements. (2) Interest in the SMM properties of 5f-metal complexes, which stems from the unique characteristics of the An ions relative to 3d and 4f metal ions. The great radial expansion of the 5f over the 4f orbitals gives rise to larger potential for covalent bonding with the ligands when compared to that of the Ln^III^ ions, and this can result in a stronger magnetic exchange interaction; also, the spin-orbit coupling of the actinides is larger than that of the 4f metal ions. These two characteristics are particularly attractive for the development of improved SMMs. (3) Almost all of the previously described uranyl(VI) chemistry had been studied with the goal of understanding the fundamental electronic structure and reactivity of the cation and its behavior in the environment. Now, as a variety of new oxo chemistry with Schiff bases is developed, there should be opportunities for such reactions both stoichiometric and catalytic. (4) The further development of An chemistry with redox-active, i.e., non-innocent, Schiff-base ligands.

We close this effort by pointing out that it is hoped that the present review will be proven useful for inorganic chemists who are already work in the area of An chemistry with Schiff-base ligands or who enter into this exciting area. We shall be happy if the readers enjoy reading the review as we enjoyed writing it.

## Figures and Tables

**Figure 1 ijms-21-00555-f001:**
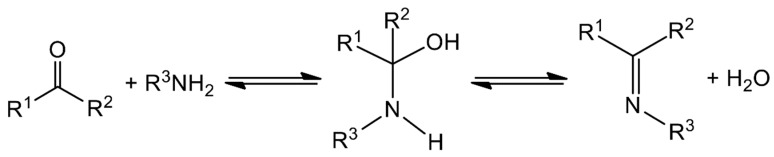
General route for the synthesis of Schiff bases.

**Figure 2 ijms-21-00555-f002:**
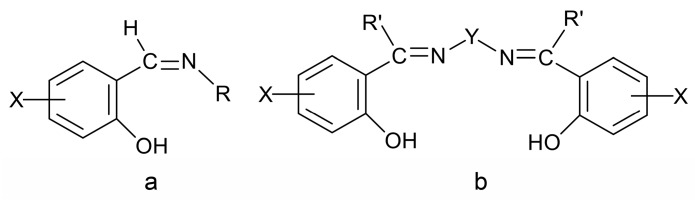
Two families of Schiff bases (**a**,**b**) that have played a significant role in the development of the coordination chemistry of these ligands and the renaissance of inorganic chemistry after the second world war. X, R, R′ = various groups (including H) and Y = (CH_2_)_x_, C_6_H_4_, etc. In a more general sense, X, R, R′ and Y can contain donor groups, thereby increasing the denticity of the Schiff bases.

**Figure 3 ijms-21-00555-f003:**
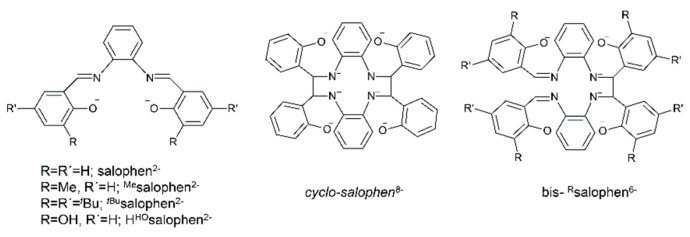
The anionic ligands that are present in complexes [Nd(salophen)X] (X = I, CF_3_SO_3_), [Nd_2_(*cyclo*-salophen)(THF)_2_]^2−^ and [Ln(bis-^R^salophen)]^3−^ reported in the text. Strictly speaking, the *cyclo*-salphen^8−^ ligand is not a Schiff base. The ligand H_2_^HO^salophen^2−^ ligand (vide infra) is also shown.

**Figure 4 ijms-21-00555-f004:**
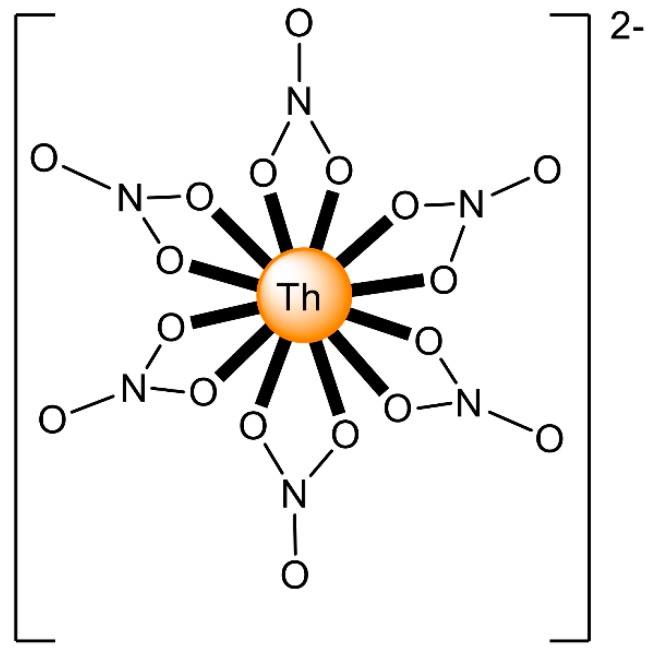
Schematic drawing of the molecular structure of [Th^IV^(NO_3_)_6_]^2−^; all the nitrato groups are bidentate chelating and the Th^IV^ center has 12-coordinates. Coordination bonds are drawn with bold lines.

**Figure 5 ijms-21-00555-f005:**
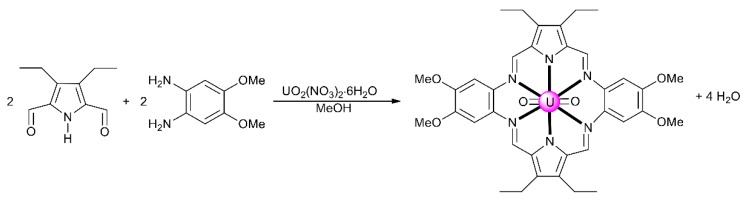
The [2 + 2] template reaction between 3,4-diethylpyrrole-2,5-dicarbaldehyde and 4,5-diamino-1,2-dimethoxybenzene to give a hexagonal bipyramidal complex with a hexadentate Schiff base as ligand.

**Figure 6 ijms-21-00555-f006:**
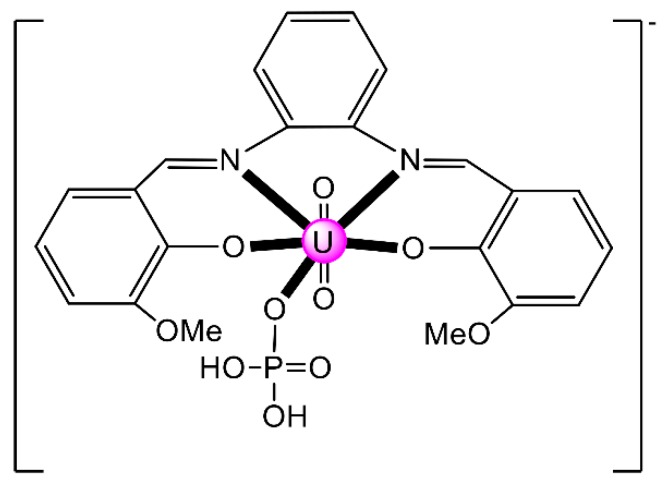
Schematic structure of the pentagonal bipyramidal anion in (*^n^*Bu_4_N)[UO_2_(^MeO^salophen)(H_2_PO_4_)]. Coordination bonds (excluding the bonds to the uranyl oxygen atoms) are drawn with bold lines.

**Figure 7 ijms-21-00555-f007:**
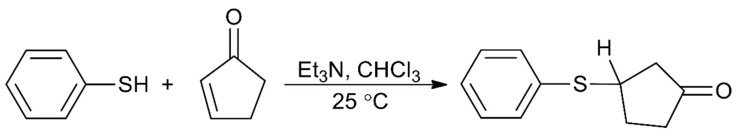
The reaction between thiophenol and 2-cyclopentene-1-one which is catalyzed by complexes [UO_2_(salophen)] and [UO_2_(^Ar^salophen)].

**Figure 8 ijms-21-00555-f008:**
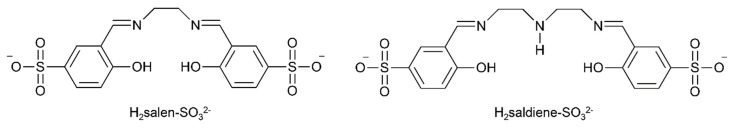
Hydrophilic polydentate Schiff-base ligands (available in the form of their disodium salts) which can selectively complex the uranyl(VI) cation and retain it in the aqueous phase; these ligands are not complexed to the same extent with trivalent Ln ions.

**Figure 9 ijms-21-00555-f009:**
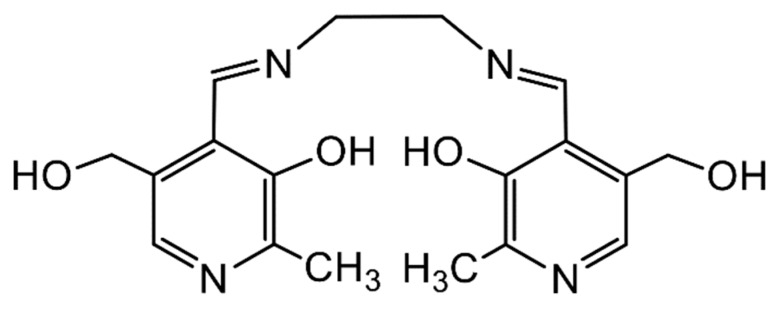
The Schiff-base ligand *N*,*N*′-bis(pyridoxylideneimine)ethylene that forms a stable chelate with Th(IV); in the complex, the donor atoms of each dianionic ligand are the (deprotonated) phenoxido O atoms and the imine N atoms.

**Figure 10 ijms-21-00555-f010:**
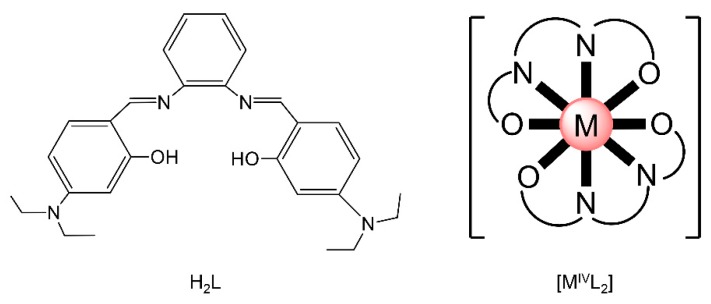
The neutral polydentate Schiff base *N*,*N*′-bis[(4,4′-diethylamino)(salicylidene]-1,2-phenylenediamine (H_2_L, **left**) and a schematic drawing of the molecular structures of the isostructural complexes [M^IV^L_2_] (**right**); M = Zr, Hf, Ce, Th, U, Pu. 
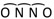
 represents the dianionic ligands, whose donor atoms are the phenoxido O atoms and the imine N atoms. Coordination bonds are drawn with bold lines.

**Figure 11 ijms-21-00555-f011:**
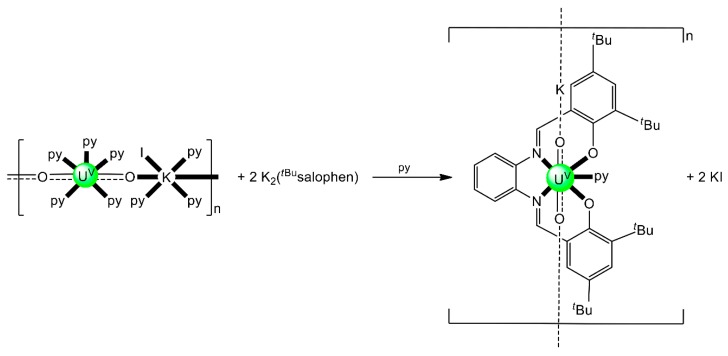
Non-reductive functionalization of the uranyl(V) compound {[UO_2_(py)_5_KI_2_(py)_2_]}_n_ with the tetradentate bulky Schiff-base ligand *^t^*^Bu^salophen^2-^; py = pyridine. Coordination bonds (except those to the pentavalent uranyl oxygens) are drawn with bold lines.

**Figure 12 ijms-21-00555-f012:**
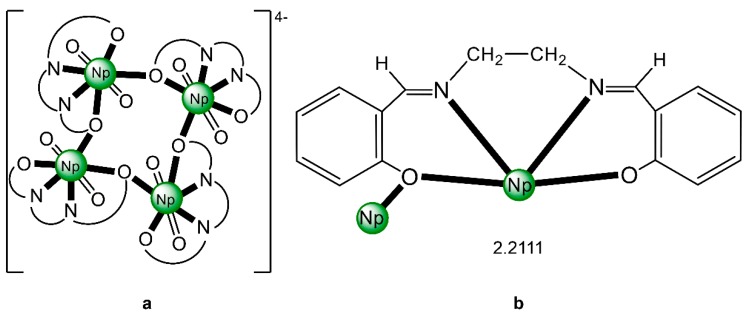
(**a**) The {Np^V^O_2_(salen)}_4_^4−^ subunit that is present in the cluster anion of **1**. (**b**) The coordination mode of salen^2−^ in **1** and the Harris notation that denotes this mode. In both diagrams, the K^I^ centers of the anion [(μ_8_-Κ)_2_{Np^V^O_2_(salen)}_4_]^2−^ have been omitted for clarity purposes. Coordination bonds (excluding the bonds to the neptunyl oxygen atoms) are drawn with bold lines.

**Figure 13 ijms-21-00555-f013:**
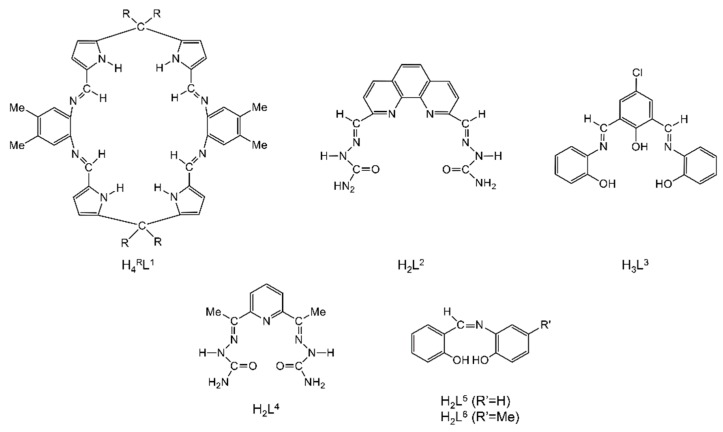
Structural formulae and abbreviation of the neutral Schiff bases which have been used for the preparation of dinuclear and oligonuclear Th(IV) and {Np^V^O_2_}^+^, and mixed Np(III){U^VI^O_2_}^2+^ complexes; for the formulae of the Th(IV) complexes, see Table 2. The carbon atoms of the imine bonds and the associated groups (H, Me) are shown with their symbols. R = Me, Et.

**Figure 14 ijms-21-00555-f014:**
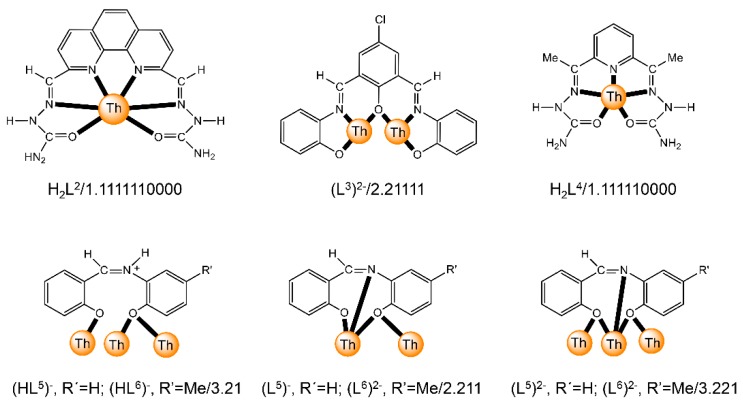
Coordination modes (Harris notation [12]) of the neutral and anionic Schiff-base ligands which have been used for the preparation of dinuclear and oligonuclear Th(IV) complexes; for the formulae of the complexes, see Table 2. The carbon atoms of the imine bond and the associated groups (H, Me) are shown with their symbols. Coordination bonds are shown with bold lines.

**Figure 15 ijms-21-00555-f015:**
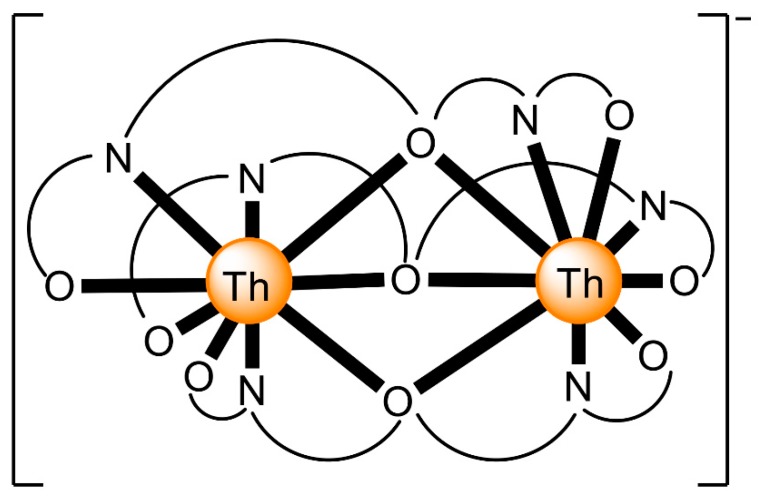
Schematic structure of the anions that are present in the crystal of **5**. 
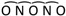
 represents the pentadentate ligand (L^3^)^3−^. Coordination bonds are drawn with bold lines.

**Figure 16 ijms-21-00555-f016:**
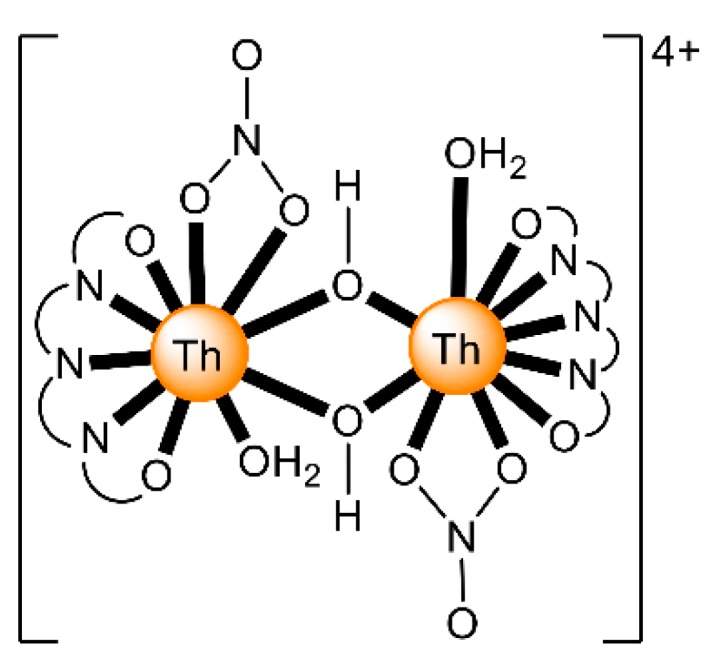
Schematic structure of the cation that is present in the crystal structure of **6**. 
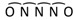
 represents the pentadentate chelating ligand H_2_L^4^. Coordination bonds are drawn with bold lines.

**Figure 17 ijms-21-00555-f017:**
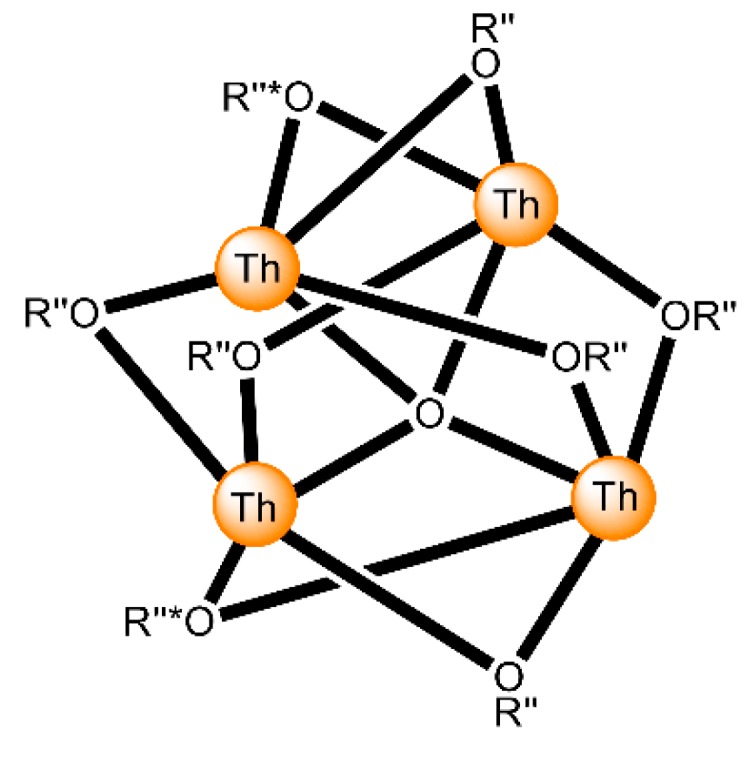
The {Th^IV^_4_(μ_4_-O)(μ-OR”)_8_ core of the molecules that are present in the structures of **7** and **8**. O is the central μ_4_ oxido group. Two of the R”O oxygen atoms (labelled by *) are the bridging atoms of the iminiumphenolate parts of the singly deprotonated ligands, while the remaining μ oxygen atoms belong to the doubly deprotonated ligands. The bonds to the Th^IV^ centers are drawn with bold lines.

**Figure 18 ijms-21-00555-f018:**
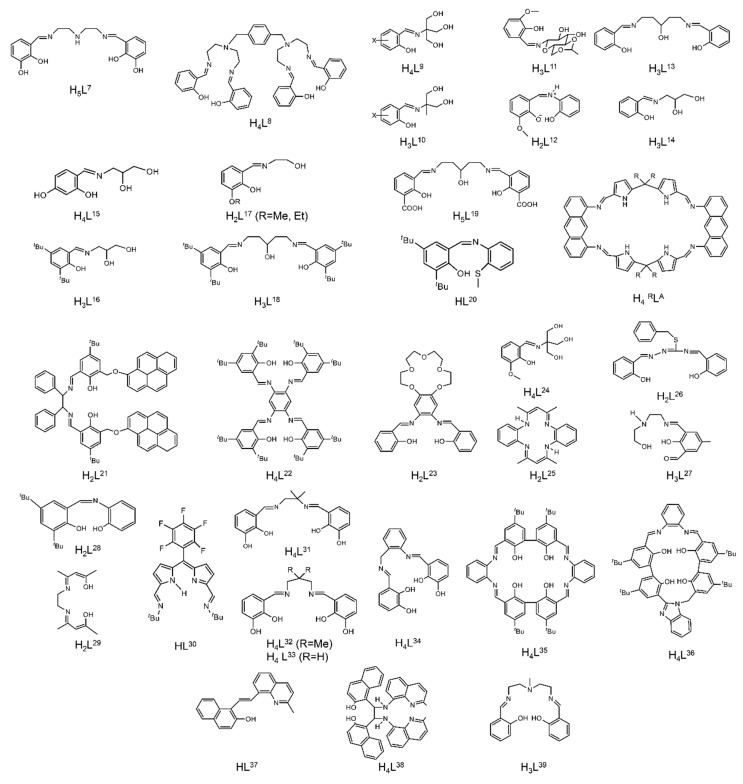
Structural formulae and abbreviations of the neutral Schiff bases which have been used for the preparation of dinuclear and oligonuclear uranium complexes; for the formulae of the complexes, see Tables 3–5. The coordination modes of many (but not all) ligands (in their anionic forms that are present in the complexes) are illustrated in Figure 19.

**Figure 19 ijms-21-00555-f019:**
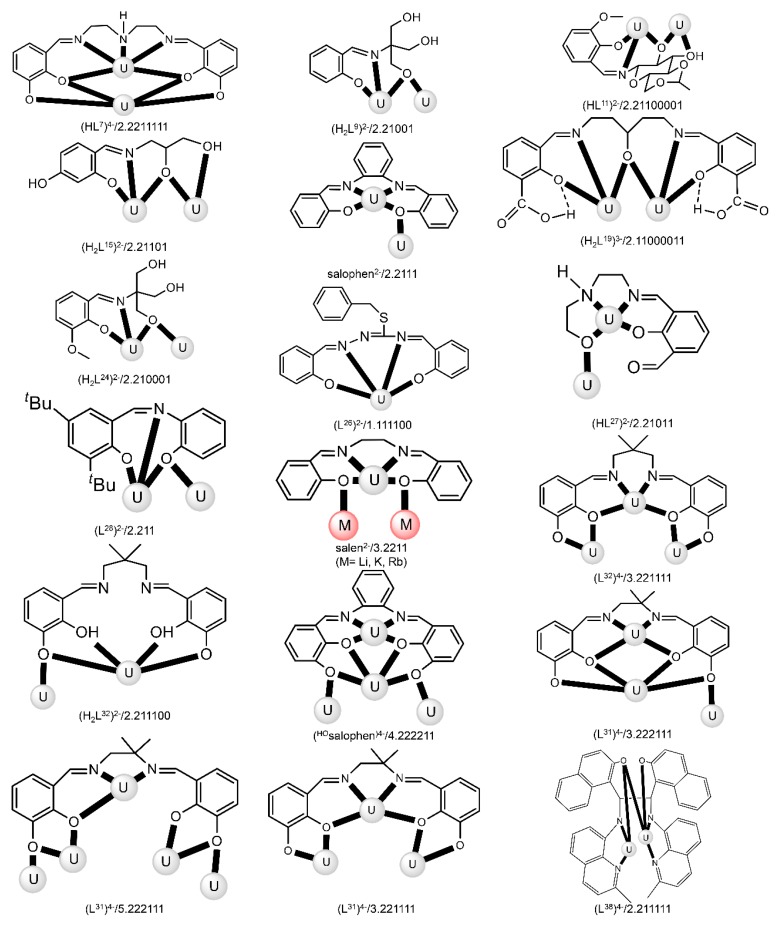
Coordination modes (Harris notation [12]) of some anionic Schiff-base ligands which have been used for the preparation of dinuclear and oligonuclear U complexes; for the formulae of the complexes, see Tables 3–5. Coordination bonds are shown with bold lines.

**Figure 20 ijms-21-00555-f020:**
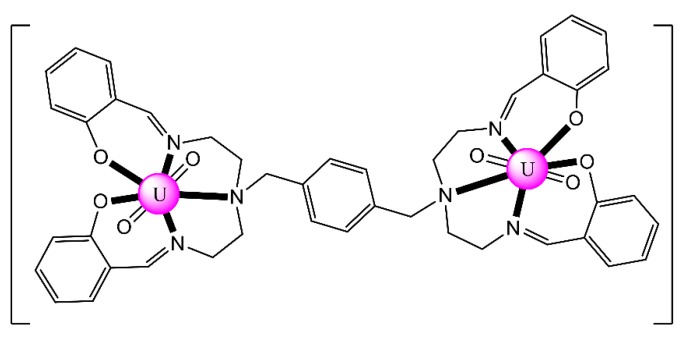
Schematic molecular structure of **10**. Coordination bonds (excluding the bonds to the uranyl oxygen atoms) are drawn with bold lines.

**Figure 21 ijms-21-00555-f021:**
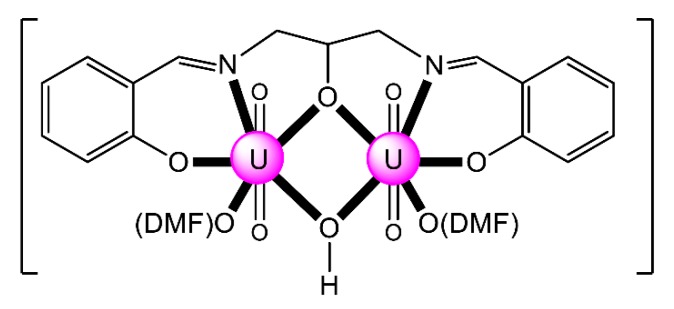
Schematic molecular structure of **15**. Coordination bonds (excluding the bonds to the uranyl oxygen atoms) are drawn with bold lines.

**Figure 22 ijms-21-00555-f022:**
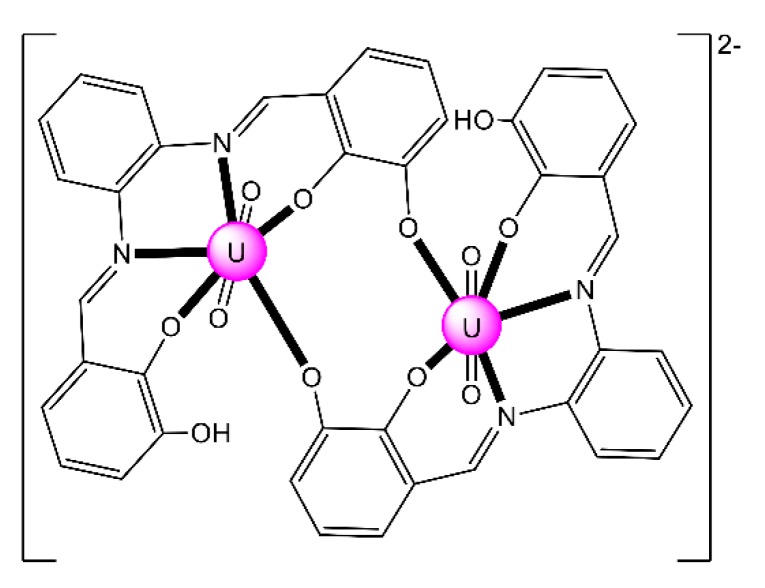
Schematic molecular structure of the dinuclear anion [(UO_2_)_2_(H^HO^salophen)_2_]^2−^ that is present in **23**. Coordination bonds (excluding the bonds to the uranyl oxygen atoms) are drawn with bold lines.

**Figure 23 ijms-21-00555-f023:**
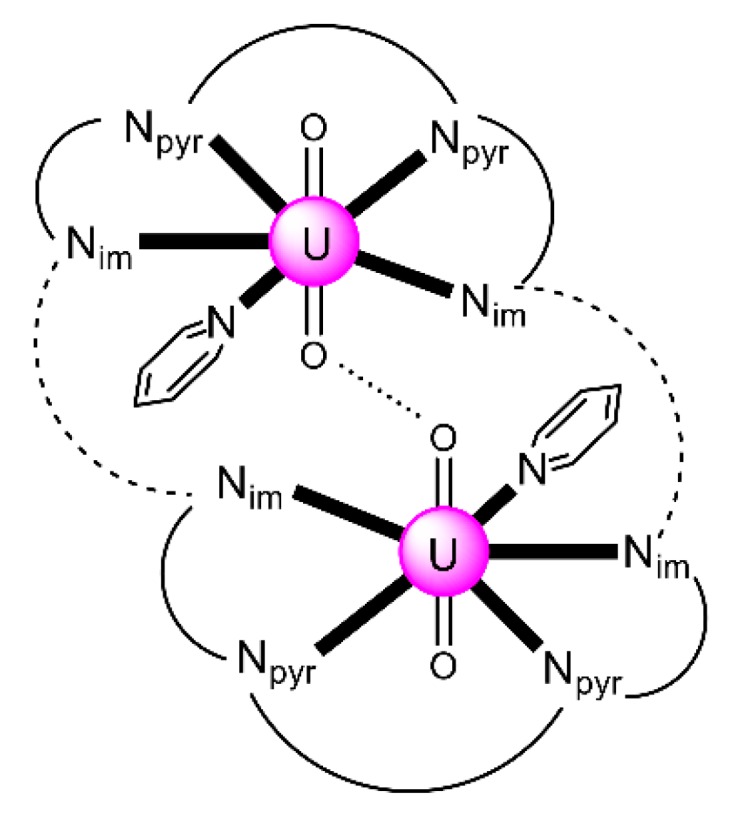
Schematic illustration of the molecular structure of **26**. N_im_ atoms represent the imine nitrogens and N_pyr_ atoms represent the deprotonated pyrrole nitrogens. The dashed lines represent the anthracenyl moieties and the solid lines the remaining parts of the (^Et^L^A^)^4−^ ligand. The dotted line represents the short oxido⋯oxido separation within the molecular cleft. Coordination bonds (excluding the bonds to the uranyl oxygen atoms) are drawn with bold lines.

**Figure 24 ijms-21-00555-f024:**
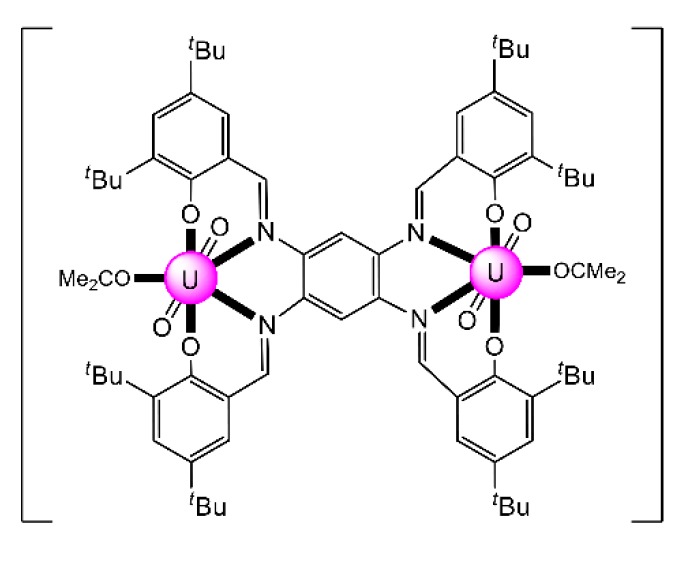
Schematic molecular structure of **31**. Coordination bonds (excluding the bonds to the uranyl oxygen atoms) are drawn with bold lines.

**Figure 25 ijms-21-00555-f025:**
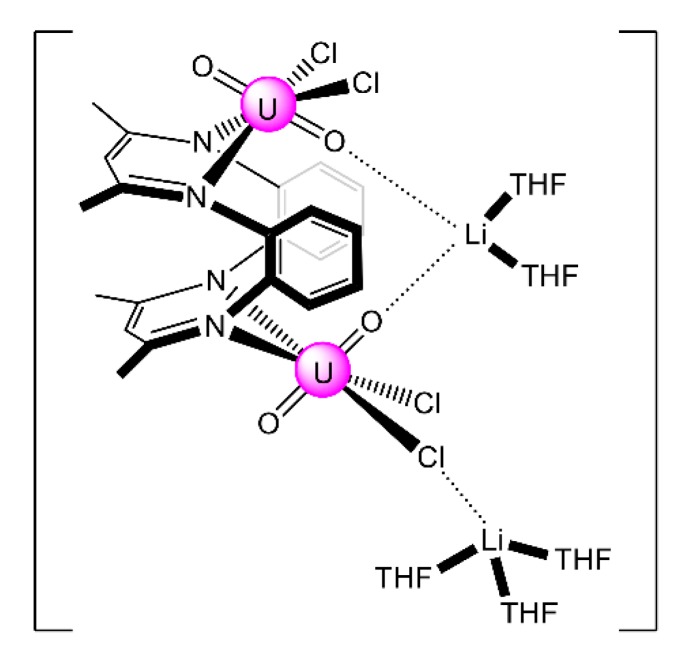
Schematic molecular structure of **36**. Coordination bonds (excluding the bonds to the uranyl oxygen atoms) are drawn with bold lines.

**Figure 26 ijms-21-00555-f026:**
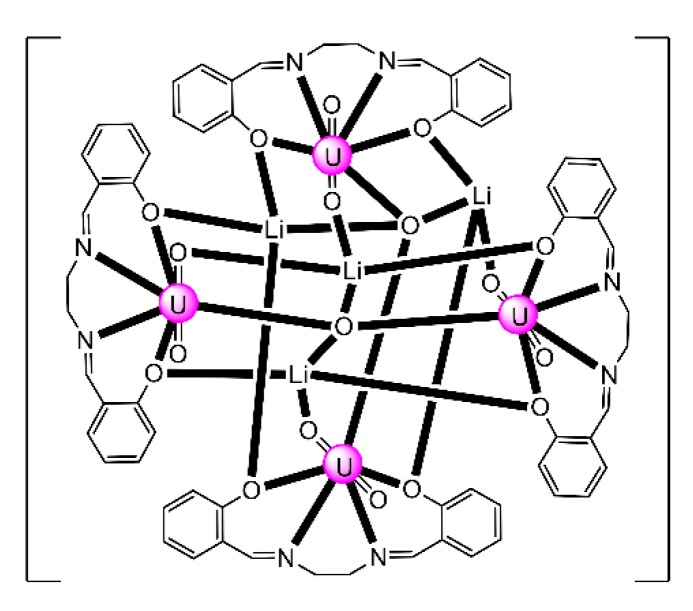
Schematic molecular structure of **41**. Coordination bonds (excluding the bonds to the uranyl oxygen atoms) and interactions are drawn with solid lines.

**Figure 27 ijms-21-00555-f027:**
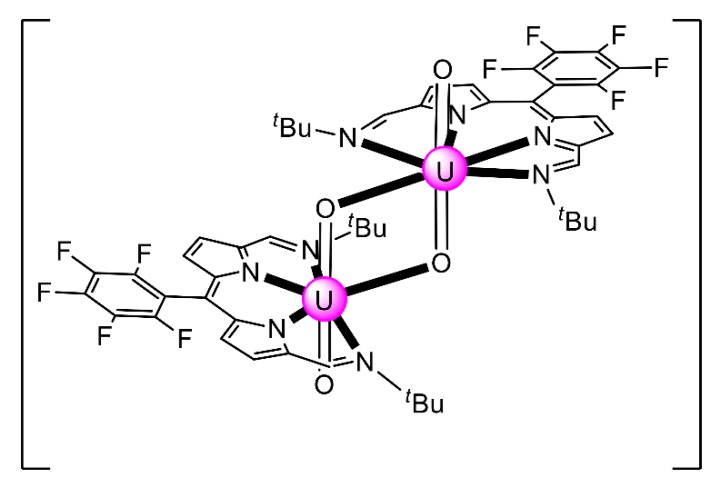
Schematic molecular structure of **52**. Coordination bonds (excluding the bonds to the uranyl oxygen atoms) or interactions are drawn with solid lines.

**Figure 28 ijms-21-00555-f028:**
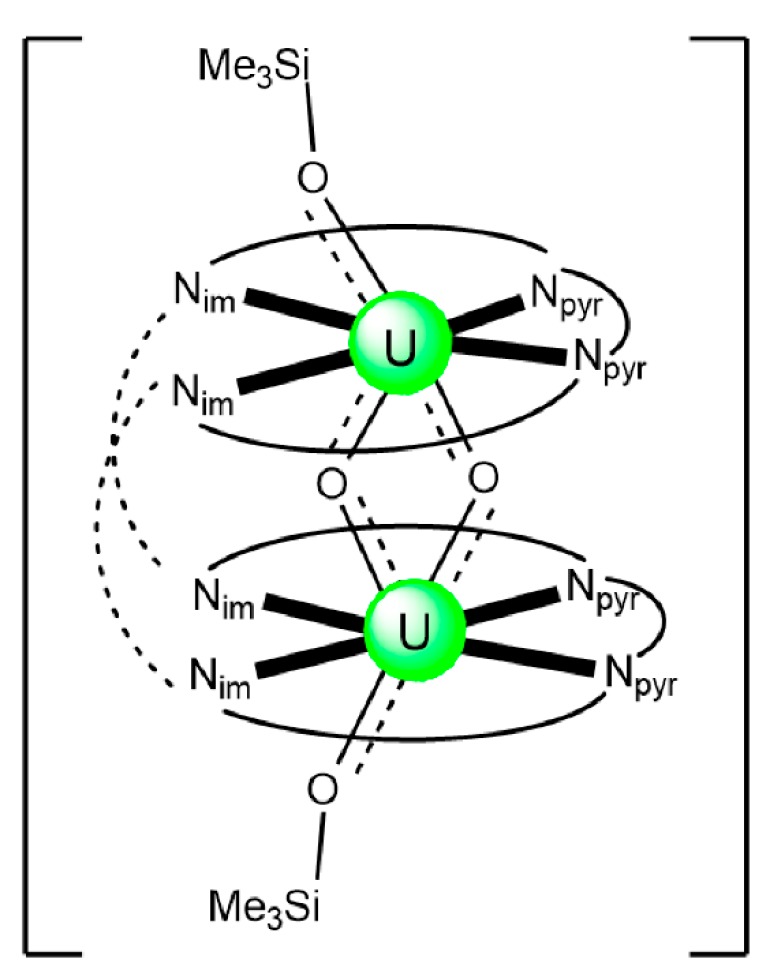
Schematic illustration of the molecular structure of **48**. N_im_ represents an imine nitrogen atom and N_pyr_ represents a deprotonated pyrrole nitrogen atom. The dashed lines represent the dimethyl phenyl rings and the solid lines the remaining parts of the (^Me^L^1^)^4−^ ligands. Coordination bonds (except those to the pentavalent uranyl oxygens) are drawn with bold lines.

**Figure 29 ijms-21-00555-f029:**
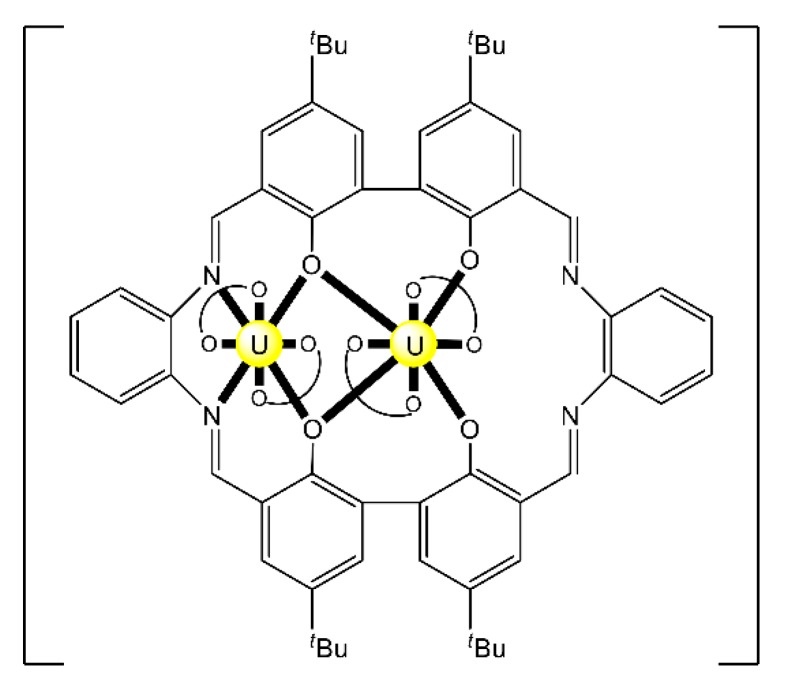
Schematic molecular structure of **66**. $\overset{⏜}{O\;O}$ represents the acac^–^ ligand. Coordination bonds are drawn with bold lines.

**Figure 30 ijms-21-00555-f030:**
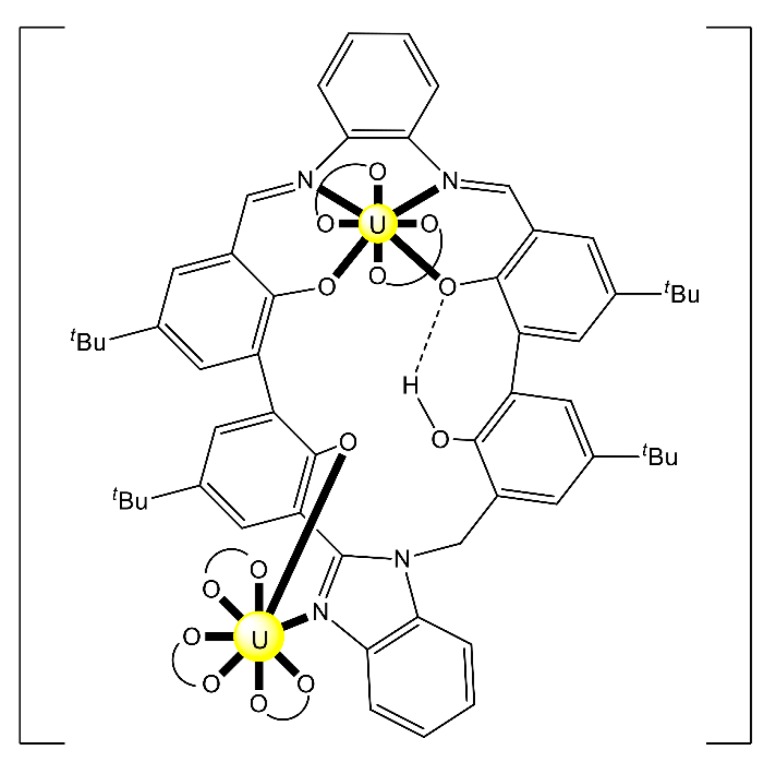
Schematic molecular structure of **67**. $\overset{⏜}{O\;O}$ represents the acac^−^ ligand. Coordination bonds are drawn with bold lines.

**Figure 31 ijms-21-00555-f031:**
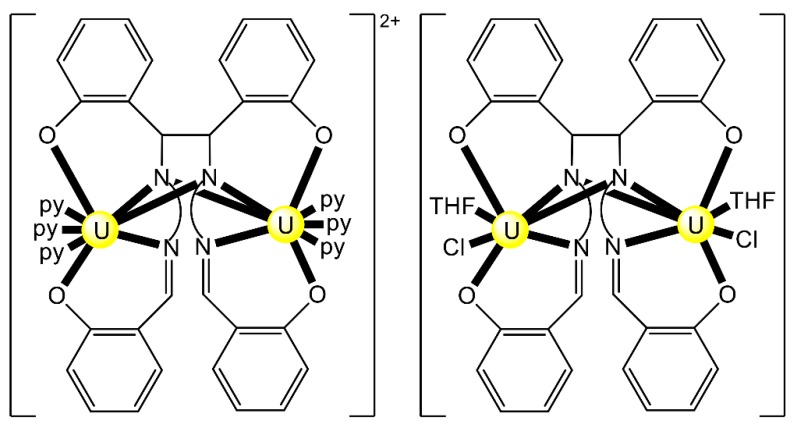
Schematic structures of the cation that is present in **70** (**left**) and the molecule that is present in (**71**) (**right**). The curves $\overset{⏜}{r}$epresent the two phenyl rings of the octadentate, hexa-anionic Schiff-base ligand. Coordination bonds are drawn with solid lines.

**Figure 32 ijms-21-00555-f032:**
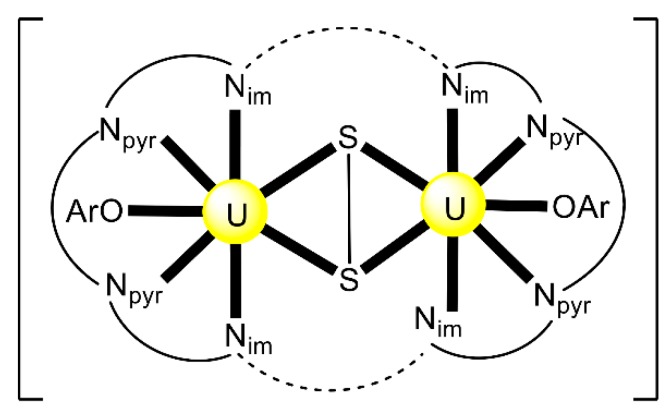
Schematic illustration of the molecular structure of **72**. N_im_ atoms represent the imine nitrogens and N_pyr_ atoms represent the deprotonated pyrrole nitrogens of the 2.11111111 (^Et^L^A^)^4−^ ligand. The dashed lines represent the anthracenyl moieties and the solid curves the remaining parts of the ligands. Coordination bonds are drawn with bold lines.

**Figure 33 ijms-21-00555-f033:**
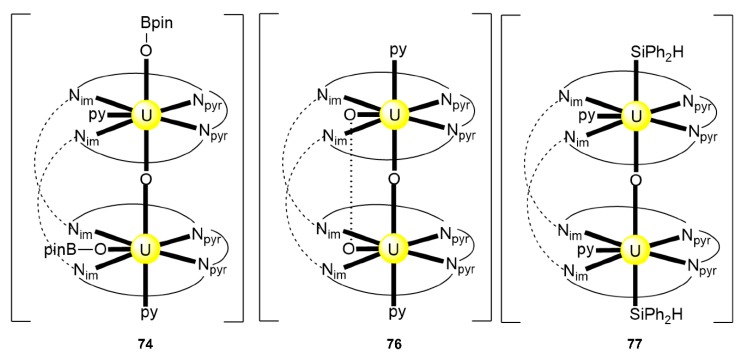
Schematic illustration of the molecular structures of **74**, **76** and **77**. N_im_ atoms represent the imine nitrogens and N_pyr_ atoms represent the deprotonated pyrrole nitrogens of the 2.11111111 (^Et^L^A^)^4−^ ligand. The dashed lines represent the anthracenyl moieties and the solid curves the remaining parts of the ligand. The dotted line in **76** represents the carbon backbone of the catecholato(-2) ligand. Coordination bonds are drawn with bold lines.

**Figure 34 ijms-21-00555-f034:**
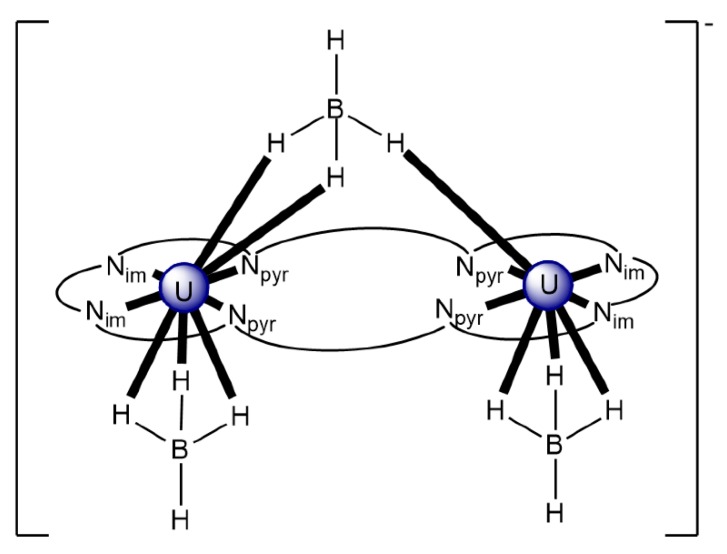
Schematic illustration of the anion that is present in the stricture of **78**. N_im_ and N_pyr_ represent the imine and pyrrolido nitrogen atoms, respectively, of the 2.11111111 (^Me^L^1^)^4−^ ligand. The solid curves represent the remaining parts of the ligand. Coordination bonds are drawn with bold lines.

**Figure 35 ijms-21-00555-f035:**
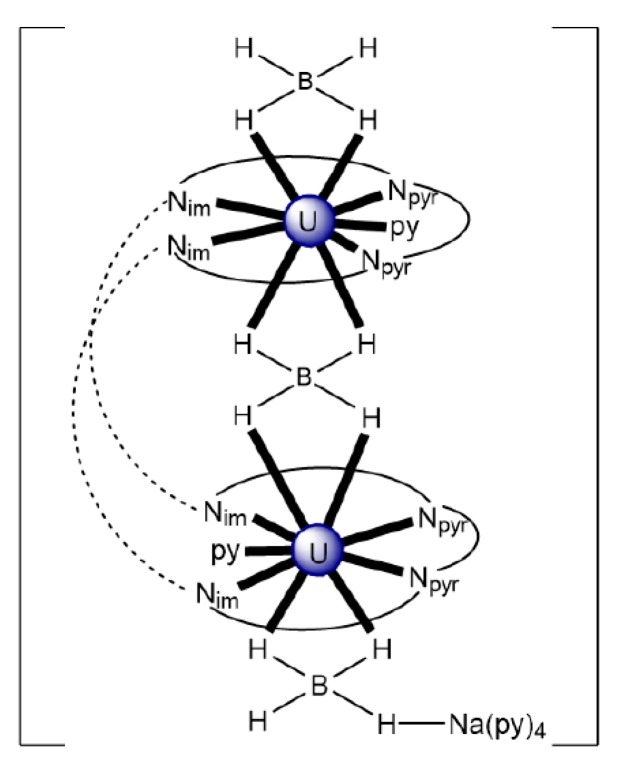
Schematic illustration of the molecular structure of **79**. N_im_ and N_pyr_ represent the imine and pyrrolido nitrogen atoms, respectively, of the 2.11111111 (^Et^L^A^)^4−^ ligand. The dashed lines represent the anthracenyl moieties and the solid curves the remaining parts of the ligand. Coordination bonds are drawn with bold lines.

**Figure 36 ijms-21-00555-f036:**
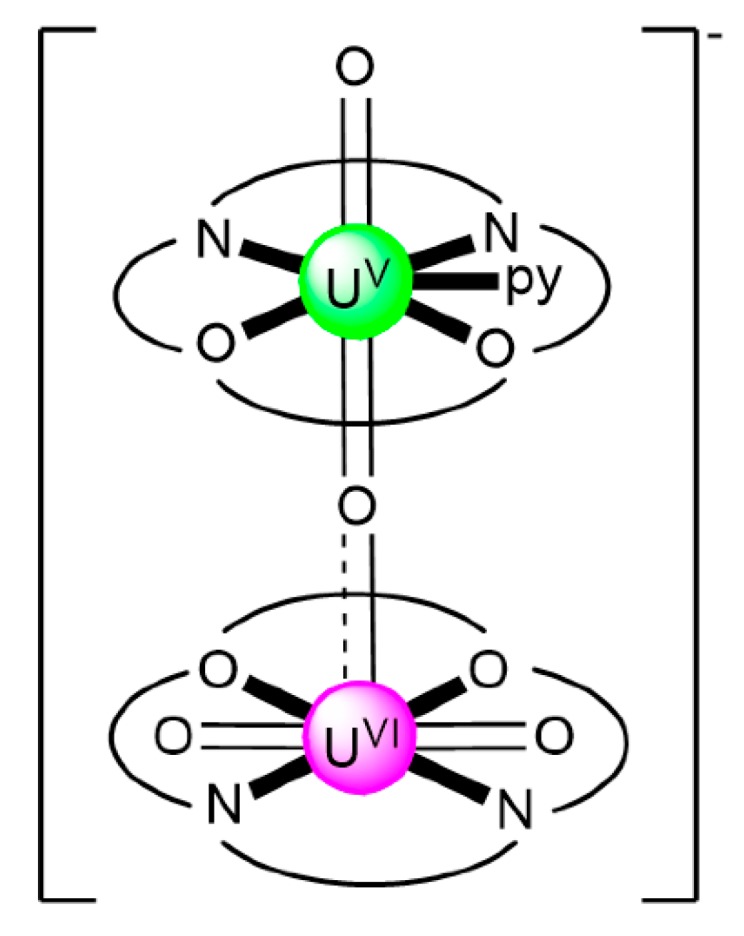
Schematic molecular structure of the dinuclear anion that is present in **83**. The solid curves represent the cation backbone of the salen^2−^ ligands. Coordination bonds (except those to the uranyl oxygen atoms) are drawn with bold lines.

**Figure 37 ijms-21-00555-f037:**
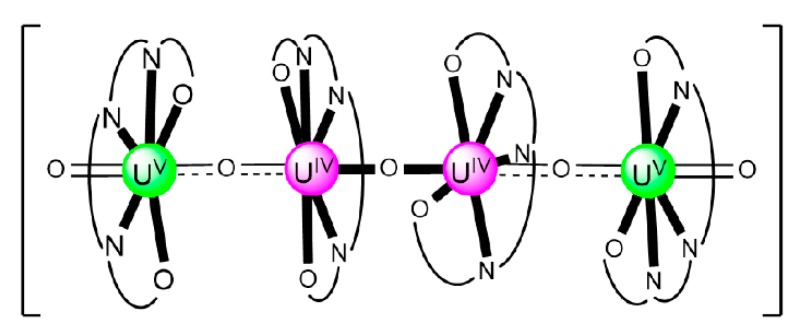
Schematic molecular structure of the tetranuclear molecule that is present in **84**. The solid curves represent the carbon backbone of the (L^39^)^2−^ ligands. Most coordination bonds are drawn with solid lines.

**Figure 38 ijms-21-00555-f038:**
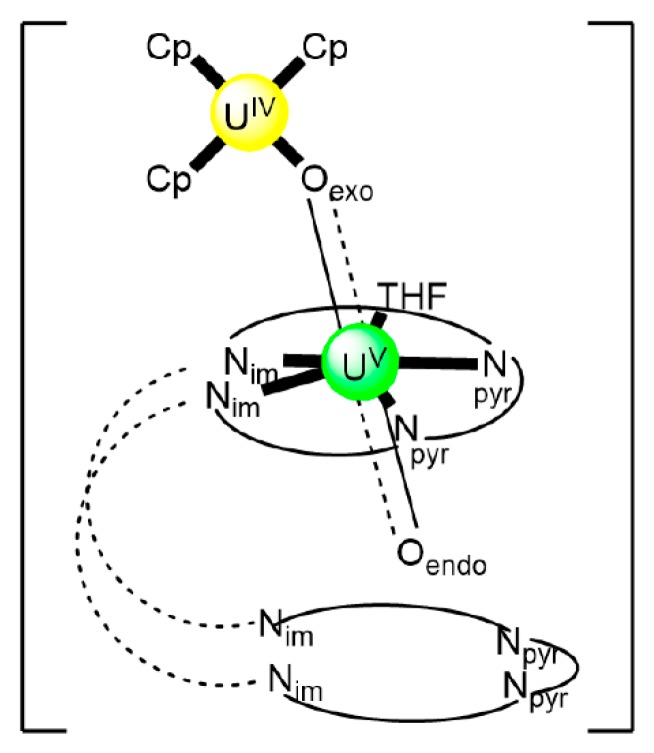
Schematic illustration of the molecular structures of **85a** and **85b.** N_im_ and N_pyr_ represent the imine and pyrrole nitrogen atoms, respectively, of the 2.11110000 (H_2_^R^L^1^)^2−^ ligand; the coordinated pyrrole nitrogen atoms are deprotonated. The dashed lines represent the aryl moieties and the solid curves the remaining parts of the ligand. Coordination bonds (except those to the pentavalent uranyl oxygens) are drawn with bold lines.

**Table 1 ijms-21-00555-t001:** Some characteristics of the actinoids.

Element	Z	Electronic Configuration in the Ground State			Radius/Å		Oxidation State ^[b]^
		An	An^3+^	An^4+^	An^3+ [a]^	An^4+ [a]^	
Ac	89	[Rn]6d^1^7s^2^	[Rn]5f^o^		1.11	0.99	**III**
Th	90	[Rn]6d^2^7s^2^	[Rn]5f^1^	[Rn]5f^o^		0.94	III, **IV**
Pa	91	[Rn]5f^2^7s^2^6d^1^	[Rn]5f^2^	[Rn]5f^1^	1.04	0.90	IV, **V**
U	92	[Rn]5f^3^7s^2^6d^1^	[Rn]5f^3^	[Rn]5f^2^	1.03	0.89	II, III, **IV**, V, **VI**
Np	93	[Rn]5f^4^7s^2^6d^1^	[Rn]5f^4^	[Rn]5f^3^	1.01	0.87	III, IV, **V**, VI, VII
Pu	94	[Rn]5f^6^7s^2^	[Rn]5f^5^	[Rn]5f^4^	1.00	0.86	III, **IV**, V, VI, VII
Am	95	[Rn]5f^7^7s^2^	[Rn]5f^6^	[Rn]5f^5^	0.98	0.85	II, **III**, IV, V, VI
Cm	96	[Rn]5f^7^7s^2^6d^1^	[Rn]5f^7^	[Rn]5f^6^	0.97	0.85	**III**, IV
Bk	97	[Rn]5f^9^7s^2^	[Rn]5f^8^	[Rn]5f^7^	0.96	0.83	**III**, IV
Cf	98	[Rn]5f^10^7s^2^	[Rn]5f^9^	[Rn]5f^8^	0.95	0.82	II, **III**, IV
Es	99	[Rn]5f^11^7s^2^	[Rn]5f^10^	[Rn]5f^9^	n.k.		II, **III**
Fm	100	[Rn]5f^12^7s^2^	[Rn]5f^11^	[Rn]5f^10^	n.k.		II, **III**
Md	101	[Rn]5f^13^7s^2^	[Rn]5f^12^	[Rn]5f^11^	n.k.		II, **III**
No	102	[Rn]5f^14^7s^2^	[Rn]5f^13^	[Rn]5f^12^	n.k.		**II**, III
Lr	103	[Rn]5f^14^7s^2^6d^1^	[Rn]5f^14^	[Rn]5f^13^	n.k.		**III**

^[a]^ For a 6-coordinate ion. ^[b]^ The most stable oxidation states are written in bold. n.k. = not known.

**Table 2 ijms-21-00555-t002:** Structurally characterized dinuclear and oligonuclear Th(IV) complexes with Schiff bases as ligands.

Complex ^[a],[b]^	Coordination Mode of the Schiff Base Ligand(s) ^[c]^	Ref.
[Th_2_O(NO_3_)_2_(H_2_L^2^)_2_(H_2_O)_2_](NO_3_)_4_ (**4**)	1.1111110000	[65]
[Mg(H_2_O)_6_][Th_2_(L^3^)_3_]_2_ (**5**)	2.21111	[66]
[Th_2_(OH)_2_(NO_3_)_2_(H_2_L^4^)_2_(H_2_O)_2_](NO_3_)_4_ (**6**)	1.1111110000	[67]
[Th_4_O(NO_3_)_2_(HL^5^)_2_(L^5^)_5_] (**7**)	3.21/(HL^5^)^−^, 2.211(L^5^)^2−^, 3.221/(L^5^)^2−^	[16]
[Th_4_O(NO_3_)_2_(HL^6^)_2_(L^6^)_5_] (**8**)	3.21/(HL^6^)^−^, 2.211(L^6^)^2−^, 3.221/(L^6^)^2−^	[16]

^[a]^ Lattice solvent molecules have been omitted. ^[b]^ For the structural formulae of the neutral Schiff-base ligands, see Figure 13. ^[c]^ For the description of the coordination modes, the Harris notation is used [12]; the modes are illustrated in Figure 14.

**Table 3 ijms-21-00555-t003:** Structurally characterized dinuclear, trinuclear and tetranuclear uranyl(VI) complexes with Schiff bases as ligands.

Complex ^[a],[b]^	Coordination Mode of the Schiff-Ligand ^[c]^	Ref.
[(UO_2_)_2_(HL^7^)(S)] (**9**) ^[d]^	2.2211111	[81]
[(UO_2_)_2_(L^8^)] (**10**)	2.1111111111	[82]
[(UO_2_)_2_(H_2_L^9^)_2_(H_2_O)_2_] (**11**)	2.21001	[83]
[(UO_2_)_2_(HL^10^)_2_(H_2_O)_2_] (**12**)	2.2101	[83]
[(UO_2_)_2_(HL^11^)_2_] (**13**)	2.11100001	[84]
[(UO_2_)_2_(L^12^)_2_(THF)_2_] (**14**)	2.2101	[85]
[(UO_2_)_2_(OH)(L^13^)(DMF)_2_] (**15**)	2.21111	[86]
[(UO_2_)_2_(HL^14^)_2_] (**16**)	2.2111	[87]
[(UO_2_)_2_(H_2_L^15^)_2_] (**17**)	2.21101	[87]
[(UO_2_)_2_(HL^16^)_2_] (**18**)	2.2111	[87]
[(UO_2_)_2_(salophen)_2_] (**19**)	2.2111	[88]
[(UO_2_)_2_(L^17^)_2_(DMF)_2_] (**20**) ^[e]^	2.2101	[89]
[(UO_2_)_2_(L^17^)_2_(DMF)_2_] (**21**) ^[f]^	2.2101	[90]
[(UO_2_)_2_(OH)(L^18^)(DMF)_2_] (**22**)	2.21111	[91]
(Et_3_NH)_2_[(UO_2_)_2_(H^HO^salophen)_2_] (**23**)	2.211011	[92]
[K_2_(UO_2_)_2_(OH)_2_(H_2_^Me^L^1^)_2_(C_6_H_6_)_2_] (**24**)	1.11110000	[93]
[(UO_2_)_2_(OH)(H_2_L^19^)(DMSO)_2_] (**25**)	2.211000011	[94]
[(UO_2_)_2_(^Et^L^A^)(py)_2_] (**26**)	2.11111111	[95]
[K_2_(UO_2_)_2_(O_2_)(^Me^L^1^)] (**27**)	2.11111111	[96]
[K_2_(UO_2_)_2_(O)(^Me^L^1^)] (**28**)	2.11111111	[96]
[(UO_2_)_2_Cl_2_(L^20^)_2_] (**29**)	1.111	[97]
(Me_4_N)[(UO_2_)_2_(OH)(L^21^)_2_] (**30**)	1.110011	[98]
[(UO_2_)_2_(L^22^)(Me_2_CO)_2_] (**31**)	2.11111111	[99]
[Li_2_(UO_2_)_2_Cl_2_(L^23^)_2_(H_2_O)_2_] (**32**)	2.111111011	[100]
[Na_2_(UO_2_)_2_Br_2_(L^23^)_2_(H_2_O)(MeOH] (**33**)	2.111111111	[100]
[(UO_2_)_2_(H_2_L^24^)_2_(DMSO)_2_] (**34**)	2.210001	[101]
[(UO_2_)_2_(L^6^)_2_(EtOH)_2_] (**35**)	2.211	[15]
[Li_2_(UO_2_)_2_Cl_4_(L^25^)(THF)_5_] (**36**)	2.1111	[102]
(Et_3_NH)[(UO_2_)_2_(O_2_CMe)(L^26^)_2_ (**37**)	1.111100	[103]
[(UO_2_)_2_(HL^27^)_2_] (**38**)	2.21011	[104]
(Et_3_NH)[(UO_2_)_3_(OH)(L^28^)_3_] (**39**)	2.211	[105]
(Et_3_NH)_2_[(UO_2_)_3_(O)(L^28^)_3_] (**40**)	2.211	[105]
[Li_4_(UO_2_)_4_(O)_2_(salen)_4_] (**41**)	3.2211	[106]

^[a]^ Lattice solvent molecules have been omitted. ^[b]^ For the structural formulae of the anionic/neutral Schiff-base ligands, see Figure 2, Figure 3, Figure 13 and Figure 18. ^[c]^ For the description of the coordination modes, the Harris notation is used [12]; the modes of the ligands are discussed in the text, and some of them are illustrated in Figure 19. ^[d]^ S = DMF, DMSO. ^[e]^ R = Me in (L^17^)^2−^. ^[f]^ R = Et in (L^17^)^2−^.

**Table 4 ijms-21-00555-t004:** Structurally characterized dinuclear and oligonuclear homovalnet uranium(V), uranium(IV) and uranium(III) complexes with Schiff bases as ligands.

Complex ^[a],[b]^	Coordination Mode of the Schiff-Base Ligand(s) ^[c]^	Ref.
[K(18C6)(py)]_2_[K_2_(U^V^O_2_)_4_(salen)_4_] (**42**)	3.2211	[107]
[K(18C6)(py)]_2_[K_2_(U^V^O_2_)_4_(L^29^)_4_] (**43**)	3.2211	[106]
[K(222)(py)]_2_[K_2_(U^V^O_2_)_4_(L^29^)_4_] (**44**)	3.2211	[106]
[K(18C6)(THF)]_2_[K_6_(U^V^O_2_)_4_(salopen)_4_I_2_(18C6)_2_]I_2_ (**45**)	3.2211	[106]
[Rb_4_(U^V^O_2_)_4_(salen)_4_(18C6)_2_] (**46**)	3.2211	[106]
[K_2_(U^V^O_2_)_2_(^Me^L^1^)] (**47**)	2.11111111	[96]
[(Me_3_SiOU^V^O)_2_(^Me^L^1^)] (**48**)	2.11111111	[108]
[Li_2_(U^V^O_2_)_2_(^Me^L^1^)(py)_3_] (**49**)	2.11111111	[109]
[Li(U^V^O_2_)(Me_3_SiOU^V^O)(^Me^L^1^)(py)_3_] (**50**)	2.11111111	[109]
[Li_2_(U^V^O_2_)_2_(H_2_^Me^L^1^)_2_(py)_2_] (**51**)	2.11111100	[110]
[(U^V^O_2_)_2_(L^30^)_2_] (**52**)	1.1111	[111]
[Rb_6_(U^V^O_2_)_6_(H_2_^Me^L^1^)_6_(py)_6_] (**53**)	2.11111100	[112]
[Cs_6_(U^V^O_2_)_6_(H_2_^Me^L^1^)_6_(py)_6_] (**54**)	2.11111100	[112]
(pyH)_3_[U^IV^_3_(O)Cl_9_(L^31^)] (**55**)	3.221111	[113]
[U^IV^_4_(O)(L^32^)_2_(H_2_L^32^)_2_(py)_2_](CF_3_SO_3_)_2_ (R = Me) (**56**)	3.221111 ^[d]^, 2.211100 ^[e]^	[113]
(pyH)_2_[U^IV^_8_(O)_4_Cl_10_(^HO^salophen)_4_] (**57**)	4.222211	[114]
[U^IV^_3_(acac)_2_(^HO^salophen)(H^HO^salophen)_2_] (**58**)	2.221111 ^[f]^, 3.221011 ^[g]^, 2.211011 ^[g]^	[115]
[U^IV^_4_(HL^34^)_4_(H_2_L^34^)_2_] (**59**)	2.211011 ^[h]^, 2.211000 ^[i]^	[115]
(pyH)_2_[U^IV^_4_Cl_6_(L^33^)_2_(H_2_L^33^)_2_] (**60**)	2.221111 ^[j]^, 3.211100 ^[k]^	[116]
[U^IV^_4_Cl_4_(L^33^)_2_(H_2_L^33^)_2_(py)_2_] (**61)**	2.221111 ^[j]^, 3.211100 ^[k]^	[116]
(pyH)_2_[U^IV^_4_Cl_6_(L^34^)_2_(H_2_L^34^)_2_][U^IV^_4_Cl_4_(L^34^)_2_(H_2_L^34^)_2_(py)_2_] (**62**)	2.221111 ^[j]^, 3.211100 ^[k]^	[116]
(pyH)_2_[U^IV^_6_Cl_10_(L^31^)_4_(py)_4_] (**63**)	3.222111, 5.222111	[116]
[U^IV^_2_Cl_4_(L^35^)] (**64**)	2.11111111	[117]
[U^IV^_2_Cl_4_(L^35^)(py)_2_] (**65**)	2.11111111	[117]
[U^IV^_2_(acac)_4_(L^35^)] (**66**)	2.22111100	[117]
[U^IV^_2_(acac)_5_(HL^36^)] (**67**)	2.11101110	[117]
[U^IV^(*cyclo*-salophen)(py)_4_] (**68**)	2.11112222	[118]
[U^IV^_2_(L^38^)_2_] (**69**) ^[l]^	2.211111	[119]
[U^IV^_2_(bis-^H^salophen)(py)_6_]I_2_ (**70**)	2.11112211	[120]
[U^IV^_2_Cl_2_(bis-^H^salophen)(THF)_2_] (**71**)	2.11112211	[120]
[U^IV^_2_(OAr)_2_(S_2_)(^Et^L^A^)] (**72**) ^[m]^	2.11111111	[121]
[U^IV^_2_(OAr)_2_(S)(^Et^L^A^)] (**73**) ^[m]^	2.11111111	[121]
[{pinBO)U^IV^OU^IV^(OBpin)}(^Et^L^A^)(py)_2_] (**74**) ^[n}^	2.11111111	[122]
[{(py)catBO}U^IV^OU^IV^(OBcat)(^Et^L^A^)(py)_2_] (**75**) ^[o}^	2.11111111	[122]
[{U^IV^OU^IV^(O_2_C_6_H_4_)})(^Et^L^A^)(py)_2_] (**76**) ^[p]^	2.11111111	[122]
[{(HPh_2_SiO)U^IV^OU^IV^(OSiPh_2_H)}(^Et^L^A^)(py)_2_] (**77**)	2.11111111	[122]
[Li(THF)][U^III^_2_(BH_4_)_3_(^Me^Li^1^)] (**78**)	2.11111111	[123]
[NaU^III^_2_(BH_4_)_3_(^Et^L^A^)(py)_6_] (**79**)	2.11111111	[123]
[NaU^III^_2_(OAr)_2_(BH_4_)(^Et^L^A^)(THF)_2_] (**80**) ^[m]^	2.11111111	[121]
[KU^III^_2_(OAr)_2_(BH_4_)(^Et^L^A^)(THF)_2_] (**81**) ^[m]^	2.11111111	[121]

^[a]^ Lattice solvent molecules have been omitted. ^[b]^ For the structural formulae of the anionic/neutral Schiff-base ligands, see Figure 2, Figure 3, Figure 13 and Figure 18. ^[c]^ For the description of the coordination modes, the Harris notation is used [12]; the modes of the ligands are discussed in the text, while some of them are illustrated in Figure 19. ^[d]^ For the (L^32^)^4−^ ligands. ^[e]^ For the (HL^32^)^2−^ ligands. ^[f]^ For the ^HO^salophen^4−^ ligands. ^[g]^ For the H^HO^salophen^3−^ ligands. ^[h]^ For the (HL^34^)^3−^ ligands. ^[i]^ For the (H_2_L^34^)^2−^ ligands. ^[j]^ For the quadruply deprotonated ligands. ^[k]^ For the doubly deprotonated ligands. ^[l]^ The (L^38^)^4−^ ligand is not a Schiff base; however, this complex was isolated from a reaction mixture containing the potassium salt of the Schiff base HL^37^. ^[m]^ Oar = OC_6_H_2_*^t^*Bu_3_-2,4,6. ^[n]^ pin = pinacolate = OC(Me)_2_C(Me)_2_O(-2); ^[o]^ cat = catecholate(-2) group; ^[p]^ C_6_H_4_O_2_ is the catecholato(-2) ligand.

**Table 5 ijms-21-00555-t005:** Structurally characterized mixed-valence uranium complexes with Schiff bases as ligands.

Complex ^[a],[b]^	Coordination Mode of the Schiff-Base Ligand ^[c]^	Ref.
[K_3_(U^VI^O_2_)(U^V^O_2_)_3_(salen)_4_(18C6)] (**82**)	3.2211	[107]
[Co^III^(Cp*)_2_][(U^VI^O_2_)(U^V^O_2_)(salen)_2_(py)] (**83**)	1.1111	[106]
[(U^V^O_2_)U^IV^_2_(O)(L^39^)_4_] (**84**)	1.11111	[124]
[(U^V^O_2_)_2_U^IV^_3_(O)_2_(*^t^*^Bu^salophen)_2_(salen)_3_] (**85**)	1.1111 ^[d]^, 2.2111 ^[e]^	[124]
[(U^V^O_2_)U^IV^(Cp)_3_(H_2_^Et^L^1^)(THF)] (**85a**)	2.11110000	[64]
[(U^V^O_2_)U^IV^(Cp)_3_(H_2_^Me^L^1^)(THF)] (**85b**)	2.11110000	[64]

^[a]^ Lattice solvent molecules have been omitted. ^[b]^ For the structural formulae of the anionic/neutral Schiff-base ligands, see Figure 2, Figure 3, Figure 13 and Figure 18. ^[c]^ For the description of the coordination modes, the Harris notation [12] is used; the modes of the ligands are discussed in the text, while some of them are illustrated in Figure 19. ^[d]^ For the (*^t^*^Bu^salophen)^2−^ ligands and one salen^2−^ ligand. ^[e]^ For two of the salen^2−^ ligands.

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
