# Peer review of "Oligonuclear Actinoid Complexes with Schiff Bases as Ligands—Older Achievements and Recent Progress"

_ijms, 2020, doi:10.3390/ijms21020555_

Round 1
Reviewer 1 Report
This is an interesting piece of work on the coordination chemistry of Schiff bases with 5f-metal ions in the context of oligonuclear complexes. The combination of a Schiff-base with f-elements appears to be a promising research area with broad applicability in various fields.
This review article is divided into eight carefully chosen sections presenting an overview of the significant advances and impressive discoveries that this emerging and challenging research field has witnessed since 2000. The 124 well-selected literature references provide an excellent overview of this scientific area.
The manuscript is not only well-documented but also well-organized and well-written. I will elaborate below.
In the first section, the authors justify the organization of scientific information, which seems well-thought and straightforward. Then follows a part presenting an overview of Schiff-bases. This section ingeniously recalls important fundamental aspects associated with these molecules, such as the methods of formation, the spectroscopic attributes, and coordination chemistry. The portfolio of their application is then clearly presented, encompassing many fields from organic chemistry to ligands for metal complexes.
Subsequently, a third section proposes an overview of the chemistry of actinoids, including the chemical properties, challenges, relevance, and pitfalls.
These two nicely chosen sections (2 and 3) provide to the readers as an "hors-d'œuvre" (spelling in the manuscript isn't correct), explicit basic knowledge before entering the core of the topic. The information is well-distilled; the authors go to the point while providing the necessary information. This is important enough to notice as it is not often the case in contemporary literature.
Subsequently, the authors make a smooth transition toward the core of the review by presenting in a fourth section the scientific relevance of Actinoid-Schiff Base complexes and the benefits of this ligand-Metal combination.
Then, comes a short section about rare examples of oligonuclear Schiff-base complexes that are not related to Thorium and Uranium (i.e., Pu, Np, Np/U).
The following sections 6 and 7 then discuss the elements Thorium and Uranium, respectively. It is for these two elements that most examples have been reported in the literature. These sections provide a thorough overview of the coordination tendencies and reactivities of these molecular entities.
Finally, in section 8, the authors present their final comments and some perspectives. This section provides a good summary of the current state of the research field while providing guidelines for future developments.
More general comments: It is possible to follow readily the approach taken by the authors. The scientific part is presented, explained, and discussed appropriately. The authors made a substantial effort in providing precise, concise, and well-documented work.
This review presents to the readers an excellent piece of basic knowledge in the field while smoothly introducing them to the inherent challenges.
As minor corrections, while this reviewer agrees that it can be challenging to draw coordination compounds. This is a more comfortable practice when it comes to purely organic entities. In this respect, figures 2, 3, 5, 6, 7, 8 could be improved to at least be more representative of relative bond lengths and angles between the elements. Most confusing are the drawings around several carbon atoms.
Finally, the conclusions and outlook adequately summarize the body of the work.
Author Response
As minor corrections, while this reviewer agrees that it can be challenging to draw coordination compounds. This is a more comfortable practice when it comes to purely organic entities. In this respect, figures 2, 3, 5, 6, 7, 8 could be improved to at least be more representative of relative bond lengths and angles between the elements.
The comment is correct. We improved Figures 2, 3, 5, 6, 7 and 8 to be more representative of relative bond lengths and angles between the elements.
"hors-d'œuvre" (spelling in the manuscript isn't correct)
We have corrected the mistake
Reviewer 2 Report
In the suggested review, the authors report the synthetic chemistry, reactivity studies, structures and some properties of actinoid (Np, Th, U) complexes with Schiff-base ligands. Schiff bases are important compounds in the filed of organic chemistry, providing many interesting and intriguing properties, and the transition-metal coordination complexes with Schiff-base ligands are numerous and well investigated. Since the most of the actinoid elements are radioactive, the corresponding metal complexes are not diverse. Most of the reported synthetic routes and strategies in the proposed review refer to the uranium`s chemistry, since the number of the neptunium and thorium complexes with Schiff bases as ligands is very limited and not investigated enough for the moment. However, it is nice and extensive literature search covering the proposed thematic. In contrast to many reviews published by different authors, proposed manuscript has interesting details about the elements (e.g. historical facts and legends), exceeding the area of severe and strict chemistry, making the text more vivid.
There are some minor changes that should be made before accepting the manuscript to be published:
Try to remove ! from the introduction part. This kind of innotation is not needed.
In all the figures change coordination bonds from bold lines to dashed lines.
When reporting TOF values (for instance on the page 9 or 27), the time of the reaction should be reported, TOF at which minute for example.
Sal is the salicylate(2-), and not -2 ligand (page 22).
Author Response
Try to remove ! from the introduction part. This kind of annotation is not needed.
We have removed this kind of punctuation mark from the introduction part.
In all the figures change coordination bonds from bold lines to dashed lines.
We fully respect the Reviewer’s comment. However, we prefer to leave the coordination bonds with bold lines. Thus, emphasis is given on the coordination environment about the actinoid ion which is a main feature of the discussion. In addition, we avoid (a) confusion with the dashed lines used to indicate the coordination bonds to the pentavalent uranyl oxygen atoms, e.g. in Figures 11, 28, 36, 37, 38, and (b) confusion with the dashed lines used to indicate connectivities within some ligands’ frameworks, e.g. in Figures 28, 32, 33, 35, 38. Thus, we are asking Referee’s 2 and your indulgence to retain the coordination bonds with bold lines.
When reporting TOF values (for instance on the page 9 or 27), the time of the reaction should be reported, TOF at which minute for example.
The comment is absolutely correct. We have added the reaction times (as indicated in the original literature) for both cases.
Sal is the salicylate(2-), and not -2 ligand (page 22).
The comment is correct. We have corrected the mistake in the revised version of the ms.